# Protein Representation Learning by Geometric Structure Pretraining

**Zuobai Zhang**[1,2], **Minghao Xu**[1,2], **Arian Jamasb**[3],
**Vijil Chenthamarakshan**[4], **Aurélie Lozano**[4], **Payel Das**[4], **Jian Tang**[1,5,6]
Mila - Québec AI Institute[1], Université de Montréal[2], University of Cambridge[3]
IBM Research[4], HEC Montréal[5], CIFAR AI Chair[6]
`{zuobai.zhang, minghao.xu}@mila.quebec, arj39@cam.ac.uk`
`{ecvijil,aclozano,daspa}@us.ibm.com, jian.tang@hec.ca`

## ABSTRACT

Learning effective protein representations is critical in a variety of tasks in biology such as predicting protein function or structure. Existing approaches usually pretrain protein language models on a large number of unlabeled amino acid sequences and then finetune the models with some labeled data in downstream tasks. Despite the effectiveness of sequence-based approaches, the power of pretraining on known protein structures, which are available in smaller numbers only, has not been explored for protein property prediction, though protein structures are known to be determinants of protein function. In this paper, we propose to pretrain protein representations according to their 3D structures. We first present a simple yet effective encoder to learn the geometric features of a protein. We pretrain the protein graph encoder by leveraging multiview contrastive learning and different self-prediction tasks. Experimental results on both function prediction and fold classification tasks show that our proposed pretraining methods outperform or are on par with the state-of-the-art sequence-based methods, while using much less pretraining data. Our implementation is available at `https://github.com/DeepGraphLearning/GearNet`.

## 1 INTRODUCTION

Proteins are workhorses of the cell and are implicated in a broad range of applications ranging from therapeutics to material. They consist of a linear chain of amino acids (residues) which fold into specific conformations. Due to the advent of low cost sequencing technologies (Ma & Johnson, 2012; Ma, 2015), in recent years a massive volume of protein sequences have been newly discovered. As functional annotation of a new protein sequence remains costly and time-consuming, accurate and efficient in silico protein function annotation methods are needed to bridge the existing sequence-function gap.

Since a large number of protein functions are governed by their folded structures, several data-driven approaches rely on learning representations of the protein structures, which then can be used for a variety of tasks such as protein design (Ingraham et al., 2019; Strokach et al., 2020; Cao et al., 2021; Jing et al., 2021), structure classification (Hermosilla et al., 2021), model quality assessment (Baldassarre et al., 2021; Derevyanko et al., 2018), and function prediction (Gligorijević et al., 2021). Due to the challenge of experimental protein structure determination, the number of reported protein structures is orders of magnitude lower than the size of datasets in other machine learning application domains. For example, there are 182K experimentally-determined structures in the Protein Data Bank (PDB) (Berman et al., 2000) *vs* 47M protein sequences in Pfam (Mistry et al., 2021) and *vs* 10M annotated images in ImageNet (Russakovsky et al., 2015).

To address this gap, recent works have leveraged the large volume of unlabeled protein sequence data to learn an effective representation of known proteins (Bepler & Berger, 2019; Rives et al., 2021; Elnaggar et al., 2021). A number of studies have pretrained protein encoders on millions of sequences via self-supervised learning. However, these methods neither explicitly capture nor leverage the available protein structural information that is known to be the determinants of protein functions.

To better utilize structural information, several structure-based protein encoders (Hermosilla et al., 2021; Hermosilla & Ropinski, 2022; Wang et al., 2022a) have been proposed. However, these models have not explicitly captured the interactions between edges, which are critical in protein structure modeling (Jumper et al., 2021). Besides, very few attempts (Hermosilla & Ropinski, 2022; Chen et al., 2022; Guo et al., 2022) have been made until recently to develop pretraining methods that exploit unlabeled 3D structures due to the scarcity of experimentally-determined protein structures. Thanks to recent advances in highly accurate deep learning-based protein structure prediction methods (Baek et al., 2021; Jumper et al., 2021), it is now possible to efficiently predict structures for a large number of protein sequences with reasonable confidence.

Motivated by this development, we develop a protein encoder pretrained on the largest possible number[1] of protein structures that is able to generalize to a variety of property prediction tasks. We propose a simple yet effective structure-based encoder called **GeomEtry-Aware Relational Graph Neural Network (GearNet)**, which encodes spatial information by adding different types of sequential or structural edges and then performs relational message passing on protein residue graphs. Inspired by the design of triangle attention in Evoformer (Jumper et al., 2021), we propose a *sparse edge message passing* mechanism to enhance the protein structure encoder, which is the first attempt to incorporate edge-level message passing on GNNs for protein structure encoding.

We further introduce a geometric pretraining method to learn the protein structure encoder based on the popular contrastive learning framework (Chen et al., 2020). We propose novel augmentation functions to discover biologically correlated protein substructures that co-occur in proteins (Ponting & Russell, 2002) and aim to maximize the similarity between the learned representations of substructures from the same protein, while minimizing the similarity between those from different proteins. Simultaneously, we propose a suite of straightforward baselines based on self-prediction (Devlin et al., 2018). These pretraining tasks perform masked prediction of different geometric or physico-chemical attributes, such as residue types, Euclidean distances, angles and dihedral angles. Through extensively benchmarking these pretraining techniques on diverse downstream property prediction tasks, we set up a solid starting point for pretraining protein structure representations.

Extensive experiments on several benchmarks, including Enzyme Commission number prediction (Gligorijević et al., 2021), Gene Ontology term prediction (Gligorijević et al., 2021), fold classification (Hou et al., 2018) and reaction classification (Hermosilla et al., 2021) verify our GearNet augmented with edge message passing can consistently outperform existing protein encoders on most tasks in a supervised setting. Further, by employing the proposed pretraining method, our model trained on fewer than a million samples achieves comparable or even better results than the state-of-the-art sequence-based encoders pretrained on million- or billion-scale datasets.

## 2  RELATED WORK

Previous works seek to learn protein representations based on different modalities of proteins, including amino acid sequences (Rao et al., 2019; Elnaggar et al., 2021; Rives et al., 2021), multiple sequence alignments (MSAs) (Rao et al., 2021; Biswas et al., 2021; Meier et al., 2021) and protein structures (Hermosilla et al., 2021; Gligorijević et al., 2021; Somnath et al., 2021). These works share the common goal of learning informative protein representations that can benefit various downstream applications, like predicting protein function (Rives et al., 2021) and protein-protein interaction (Wang et al., 2019), as well as designing protein sequences (Biswas et al., 2021).

Compared with sequence-based methods, structure-based methods should be, in principle, a better solution to learning an informative protein representation, as the function of a protein is determined by its structure. This line of works seeks to encode spatial information in protein structures by 3D CNNs (Derevyanko et al., 2018) or graph neural networks (GNNs) (Gligorijević et al., 2021; Baldassarre et al., 2021; Jing et al., 2021; Wang et al., 2022a; Aykent & Xia, 2022). Among these methods, IEConv (Hermosilla et al., 2021) tries to fit the inductive bias of protein structure modeling, which introduced a graph convolution layer incorporating intrinsic and extrinsic distances between nodes. Another potential direction is to extract features from protein surfaces (Gainza et al., 2020; Sverrisson et al., 2021; Dai & Bailey-Kellogg, 2021). Somnath et al. (2021) combined the advantages of both worlds and proposed a parameter-efficient multi-scale model. Besides, there are

---

[1] We use AlphaFoldDB v1 and v2 (Varadi et al., 2021) for pretraining, which is the largest protein structure database before March, 2022.

also works that enhance pretrained sequence-based models by incorporating structural information in the pretraining stage (Bepler & Berger, 2021) or finetuning stage (Wang et al., 2022b).

Despite progress in the design of structure-based encoders, there are few works focusing on structure-based pretraining for proteins. To the best of our knowledge, the only attempt is three concurrent works (Hermosilla & Ropinski, 2022; Chen et al., 2022; Guo et al., 2022), which apply contrastive learning, self-prediction and denoising score matching methods on a small set of tasks, respectively. Compared with these existing works, our proposed encoder is conceptually simpler and more effective on many different tasks, thanks to the proposed relational graph convolutional layer and edge message passing layer, which are able to efficiently capture both the sequential and structural information. Furthermore, we introduce a contrastive learning framework with novel augmentation functions to discover substructures in different proteins and four different self-prediction tasks, which can serve as a solid starting point for enabling self-supervised learning on protein structures.

## 3  STRUCTURE-BASED PROTEIN ENCODER

Existing protein encoders are either designed for specific tasks or cumbersome for pretraining due to the dependency on computationally expensive convolutions. In contrast, here we propose a simple yet effective protein structure encoder, named *GeomEtry-Aware Relational Graph Neural Network (GearNet)*. We utilize *sparse edge message passing* to enhance the effectiveness of GearNet, which is novel and crucial in the field of protein structure modeling, whereas previous works (Hermosilla et al., 2021; Somnath et al., 2021) only consider message passing among residues or atoms.

### 3.1  GEOMETRY-AWARE RELATIONAL GRAPH NEURAL NETWORK

Given protein structures, our model aims to learn representations encoding their spatial and chemical information. These representations should be invariant under translations, rotations and reflections in 3D space. To achieve this requirement, we first construct our protein graph based on spatial features invariant under these transformations.

**Protein graph construction.** We represent the structure of a protein as a residue-level relational graph $\mathcal{G} = (\mathcal{V}, \mathcal{E}, \mathcal{R})$, where $\mathcal{V}$ and $\mathcal{E}$ denotes the set of nodes and edges respectively, and $\mathcal{R}$ is the set of edge types. We use $(i, j, r)$ to denote the edge from node $i$ to node $j$ with type $r$. We use $n$ and $m$ to denote the number of nodes and edges, respectively. In this work, each node in the protein graph represents the alpha carbon of a residue with the 3D coordinates of all nodes $\boldsymbol{x} \in \mathbb{R}^{n \times 3}$. We use $\boldsymbol{f}_i$ and $\boldsymbol{f}_{(i,j,r)}$ to denote the feature for node $i$ and edge $(i, j, r)$, respectively, in which reside types, sequential and spatial distances are considered.

Then, we add three different types of directed edges into our graphs: sequential edges, radius edges and K-nearest neighbor edges. Among these, sequential edges will be further divided into 5 types of edges based on the relative sequential distance $d \in \{-2, -1, 0, 1, 2\}$ between two end nodes, where we add sequential edges only between the nodes within the sequential distance of 2. These edge types reflect different geometric properties, which all together yield a comprehensive featurization of proteins. More details of the graph and feature construction process can be found in Appendix C.1.

**Relational graph convolutional layer.** Upon the protein graphs defined above, we utilize a GNN to derive per-residue and whole-protein representations. One simple example of GNNs is the GCN (Kipf & Welling, 2017), where messages are computed by multiplying node features with a convolutional kernel matrix shared among all edges. To increase the capacity in protein structure modeling, IEConv (Hermosilla et al., 2021) proposed to apply a learnable kernel function on edge features. In this way, $m$ different kernel matrices can be applied on different edges, which achieves good performance but induces high memory costs.

To balance model capacity and memory cost, we use a relational graph convolutional neural network (Schlichtkrull et al., 2018) to learn graph representations, where a convolutional kernel matrix is shared within each edge type and there are $|\mathcal{R}|$ different kernel matrices in total. Formally, the relational graph convolutional layer used in our model is defined as

$$\boldsymbol{h}_i^{(0)} = \boldsymbol{f}_i, \quad \boldsymbol{u}_i^{(l)} = \sigma\left(\text{BN}\left(\sum_{r \in \mathcal{R}} \boldsymbol{W}_r \sum_{j \in \mathcal{N}_r(i)} \boldsymbol{h}_j^{(l-1)}\right)\right), \quad \boldsymbol{h}_i^{(l)} = \boldsymbol{h}_i^{(l-1)} + \boldsymbol{u}_i^{(l)}. \quad (1)$$

Specifically, we use node features $\boldsymbol{f}_i$ as initial representations. Then, given the node representation $\boldsymbol{h}_i^{(l)}$ for node $i$ at the $l$-th layer, we compute updated node representation $\boldsymbol{u}_i^{(l)}$ by aggregating features from neighboring nodes $\mathcal{N}_r(i)$, where $\mathcal{N}_r(i) = \{j \in \mathcal{V} | (j, i, r) \in \mathcal{E}\}$ denotes the neighborhood of node $i$ with the edge type $r$, and $\boldsymbol{W}_r$ denotes the learnable convolutional kernel matrix for edge type $r$. Here BN denotes a batch normalization layer and we use a ReLU function as the activation $\sigma(\cdot)$. Finally, we update $\boldsymbol{h}_i^{(l)}$ with $\boldsymbol{u}_i^{(l)}$ and add a residual connection from last layer.

## 3.2 Edge Message Passing Layer

As in the literature of molecular representation learning, many geometric encoders show benefits from explicitly modeling interactions between edges. For example, DimeNet (Klicpera et al., 2020) uses a 2D spherical Fourier-Bessel basis function to represent angles between two edges and pass messages between edges. AlphaFold2 (Jumper et al., 2021) leverages the triangle attention designed for transformers to model pair representations. Inspired by this observation, we propose a variant of GearNet enhanced with an edge message passing layer, named as **GearNet-Edge**. The edge message passing layer can be seen as a sparse version of the pair representation update designed for graph neural networks. The main objective is to model the dependency between different interactions of a residue with other sequentially or spatially adjacent residues.

Formally, we first construct a relational graph $\mathcal{G}' = (\mathcal{V}', \mathcal{E}', \mathcal{R}')$ among edges, which is also known as *line graph* in the literature (Harary & Norman, 1960). Each node in the graph $\mathcal{G}'$ corresponds to an edge in the original graph. $\mathcal{G}'$ links edge $(i, j, r_1)$ in the original graph to edge $(w, k, r_2)$ if and only if $j = w$ and $i \neq k$. The type of this edge is determined by the angle between $(i, j, r_1)$ and $(w, k, r_2)$. The angular information reflects the relative position between two edges that determines the strength of their interaction. For example, edges with smaller angles point to closer directions and thus are likely to share stronger interactions. To save memory costs for computing a large number of kernel matrices, we discretize the range $[0, \pi]$ into 8 bins and use the index of the bin as the edge type.

Then, we apply a similar relational graph convolutional network on the graph $\mathcal{G}'$ to obtain the message function for each edge. Formally, the edge message passing layer is defined as

$$\boldsymbol{m}_{(i,j,r_1)}^{(0)} = \boldsymbol{f}_{(i,j,r_1)}, \quad \boldsymbol{m}_{(i,j,r_1)}^{(l)} = \sigma \left( \text{BN} \left( \sum_{r \in \mathcal{R}'} \boldsymbol{W}_r' \sum_{(w,k,r_2) \in \mathcal{N}_r'((i,j,r_1))} \boldsymbol{m}_{(w,k,r_2)}^{(l-1)} \right) \right).$$

Here we use $\boldsymbol{m}_{(i,j,r_1)}^{(l)}$ to denote the message function for edge $(i, j, r_1)$ in the $l$-th layer. Similar as Eq. (1), the message function for edge $(i, j, r_1)$ will be updated by aggregating features from its neighbors $\mathcal{N}_r'((i, j, r_1))$, where $\mathcal{N}_r'((i, j, r_1)) = \{(w, k, r_2) \in \mathcal{V}' | ((w, k, r_2), (i, j, r_1), r) \in \mathcal{E}'\}$ denotes the set of incoming edges of $(i, j, r_1)$ with relation type $r$ in graph $\mathcal{G}'$.

Finally, we replace the aggregation function Eq. (1) in the original graph with the following one:

$$\boldsymbol{u}_i^{(l)} = \sigma \left( \text{BN} \left( \sum_{r \in \mathcal{R}} \boldsymbol{W}_r \sum_{j \in \mathcal{N}_r(i)} (\boldsymbol{h}_j^{(l-1)} + \text{FC}(\boldsymbol{m}_{(j,i,r)}^{(l)})) \right) \right), \tag{2}$$

where $\text{FC}(\cdot)$ denotes a linear transformation upon the message function.

Notably, it is a novel idea to use relational message passing to model different spatial interactions among residues. In addition, to the best of our knowledge, this is one of the first works that explore edge message passing for macromolecular representation learning (see a concurrent work (Morehead et al., 2022) for protein-protein interactions). Compared with the triangle attention in AlphaFold2, our method considers angular information to model different types of interactions between edges, which are more efficient for sparse edge message passing.

**Invariance of GearNet and GearNet-Edge.** The graph construction process and input features of GearNet and GearNet-Edge only rely on features (distances and angles) invariant to translation, rotation and reflection. Besides, the message passing layers use pre-defined edge types and other 3D information invariantly. Therefore, GearNet and GearNet-Edge can achieve E(3)-invariance.

## 4 Geometric Pretraining Methods

In this section, we study how to boost protein representation learning via self-supervised pretraining on a massive collection of unlabeled protein structures. Despite the efficacy of self-supervised

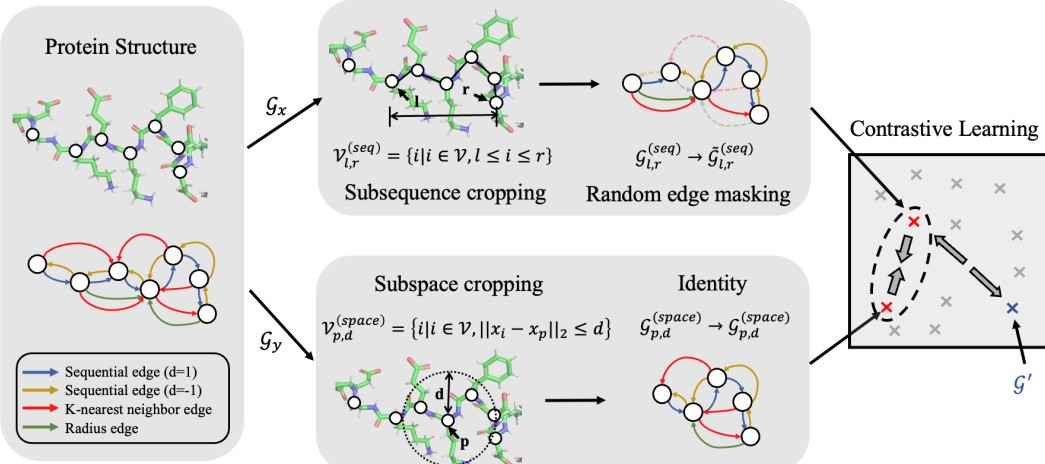

Figure 1: Demonstration of multiview contrastive learning. For each protein, we first construct the residue graph $\mathcal{G}$ based on the structural information (some edges are omitted to save space). Next, two views $\mathcal{G}_x$ and $\mathcal{G}_y$ of the protein are generated by randomly choosing the sampling scheme and noise function. For $\mathcal{G}_x$, we first extract a subsequence and then perform random edge masking with dash lines indicating masked edges. For $\mathcal{G}_y$, we perform subspace cropping and then keep the subspace graph $\mathcal{G}_{p,d}^{(\text{space})}$ unchanged. Finally, a contrastive learning loss is optimized to maximize the similarity between $\mathcal{G}_x$ and $\mathcal{G}_y$ in the latent space while minimizing its similarity with a negative sample $\mathcal{G}'$.

pretraining in many domains, applying it to protein representation learning is nontrivial due to the difficulty of capturing both biochemical and spatial information in protein structures. To address the challenge, we first introduce our multiview contrastive learning method with novel augmentation functions to discover correlated co-occurrence of protein substructures and align their representations in the latent space. Furthermore, four self-prediction baselines are also proposed for pretraining structure-based encoders.

## 4.1 MULTIVIEW CONTRASTIVE LEARNING

It is known that structural motifs within folded protein structures are biologically related (Mackenzie & Grigoryan, 2017) and protein substructures can reflect the evolution history and functions of proteins (Ponting & Russell, 2002). Inspired by recent contrastive learning methods (Chen et al., 2020; He et al., 2020), our framework aims to preserve the similarity between these correlated substructures before and after mapping to a low-dimensional latent space. Specifically, using a similarity measurement defined in the latent space, biologically-related substructures are embedded close to each other while unrelated ones are mapped far apart. Figure 1 illustrates the high-level idea.

**Constructing views that reflect protein substructures.** Given a protein graph $\mathcal{G}$, we consider two different sampling schemes for constructing views. The first one is **subsequence cropping**, which randomly samples a left residue $l$ and a right residue $r$ and takes all residues ranging from $l$ to $r$. This sampling scheme aims to capture protein domains, consecutive protein subsequences that reoccur in different proteins and indicate their functions (Ponting & Russell, 2002). However, simply sampling protein subsequences cannot fully utilize the 3D structural information in protein data. Therefore, we further introduce a **subspace cropping** scheme to discover spatially correlated structural motifs. We randomly sample a residue $p$ as the center and select all residues within a Euclidean ball with a predefined radius $d$. For the two sampling schemes, we take the corresponding subgraphs from the protein residue graph $\mathcal{G} = (\mathcal{V}, \mathcal{E}, \mathcal{R})$. In specific, the subsequence graph $\mathcal{G}_{l,r}^{(\text{seq})}$ and the subspace graph $\mathcal{G}_{p,d}^{(\text{space})}$ can be written as:

$$\mathcal{V}_{l,r}^{(\text{seq})} = \{i | i \in \mathcal{V}, l \leq i \leq r\}, \quad \mathcal{E}_{l,r}^{(\text{seq})} = \{(i,j,r) | (i,j,r) \in \mathcal{E}, i \in \mathcal{V}_{l,r}^{(\text{seq})}, j \in \mathcal{V}_{l,r}^{(\text{seq})}\},$$

$$\mathcal{V}_{p,d}^{(\text{space})} = \{i | i \in \mathcal{V}, \|\boldsymbol{x}_i - \boldsymbol{x}_p\|_2 \leq d\}, \quad \mathcal{E}_{p,d}^{(\text{space})} = \{(i,j,r) | (i,j,r) \in \mathcal{E}, i \in \mathcal{V}_{p,d}^{(\text{space})}, j \in \mathcal{V}_{p,d}^{(\text{space})}\},$$

where $\mathcal{G}_{l,r}^{(\mathrm{seq})} = (\mathcal{V}_{l,r}^{(\mathrm{seq})}, \mathcal{E}_{l,r}^{(\mathrm{seq})}, \mathcal{R})$ and $\mathcal{G}_{p,d}^{(\mathrm{space})} = (\mathcal{V}_{p,d}^{(\mathrm{space})}, \mathcal{E}_{p,d}^{(\mathrm{space})}, \mathcal{R})$.

After sampling the substructures, following the common practice in self-supervised learning (Chen et al., 2020), we apply a noise function to generate more diverse views and thus benefit the learned representations. Here we consider two noise functions: **identity** that applies no transformation and **random edge masking** that randomly masks each edge with a fixed probability 0.15.

**Contrastive learning.** We follow SimCLR (Chen et al., 2020) to optimize a contrastive loss function and thus maximize the mutual information between these biologically correlated views. For each protein $\mathcal{G}$, we sample two views $\mathcal{G}_x$ and $\mathcal{G}_y$ by first randomly choosing one sampling scheme for extracting substructures and then randomly selecting one of the two noise functions with equal probability. We compute the graph representations $\boldsymbol{h}_x$ and $\boldsymbol{h}_y$ of two views using our structure-based encoder. Then, a two-layer MLP projection head is applied to map the representations to a lower-dimensional space, denoted as $\boldsymbol{z}_x$ and $\boldsymbol{z}_y$. Finally, an InfoNCE loss function is defined by distinguishing views from the same or different proteins using their similarities (Oord et al., 2018). For a positive pair $x$ and $y$, we treat views from other proteins in the same mini-batch as negative pairs. Mathematically, the loss function for a positive pair of views $x$ and $y$ can be written as:

$$\mathcal{L}_{x,y} = -\log \frac{\exp(\mathrm{sim}(\boldsymbol{z}_x, \boldsymbol{z}_y)/\tau)}{\sum_{k=1}^{2B} \mathbb{1}_{[k \neq x]} \exp(\mathrm{sim}(\boldsymbol{z}_y, \boldsymbol{z}_k)/\tau)}, \quad (3)$$

where $B, \tau$ denotes the batch size and temperature, $\mathbb{1}_{[k \neq x]} \in \{0, 1\}$ is an indicator function that is equal to 1 *iff* $k \neq x$. The function $\mathrm{sim}(\boldsymbol{u}, \boldsymbol{v})$ is defined by the cosine similarity between $\boldsymbol{u}$ and $\boldsymbol{v}$.

**Sizes of sampled substructures.** The design of our random sampling scheme aims to extract biologically meaningful substructures for contrastive learning. To attain this goal, it is of critical importance to determine the sampling length $r - l$ for subsequence cropping and radius $d$ for subspace cropping. On the one hand, we need sufficiently long subsequences and large subspaces to ensure that the sampled substructures can meaningfully reflect the whole protein structure and thus share high mutual information with each other. On the other hand, if the sampled substructures are too large, the generated views will be so similar that the contrastive learning problem becomes trivial. Furthermore, large substructures limit the batch size used in contrastive learning, which has been shown to be harmful for the performance (Chen et al., 2020).

To study the effect of sizes of sampled substructures, we show experimental results on Enzyme Commission (*abbr.* EC, details in Sec. 5.1) using Multiview Contrast with the subsequence and substructure cropping function, respectively. To prevent the influence of noise functions, we only consider identity transformation in both settings. The results are plotted in Figure 2. For both subsequence and subspace cropping, as the sampled substructures become larger, the results first rise and then drop from a certain threshold. This phenomenon agrees with our analysis above. In practice, we set 50 residues as the length for subsequence and 15 as the radius for subspace cropping, which are large enough to capture most structural motifs according to statistics in previous studies (Tateno et al., 1997).

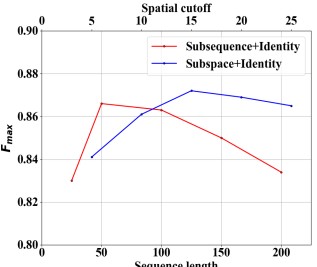

Figure 2: $F_{\max}$ on EC *v.s.* cropping sequence length (# residues) for Subsequence Cropping and spatial cutoff (Å) for Subspace Cropping.

In Appendix J, we visualize the representations of the pretrained model with this method and assign familial labels based on domain annotations. We can observe clear separation based on the familial classification of the proteins in the dataset, which proves the effectiveness of our pretraining method.

## 4.2 STRAIGHTFORWARD BASELINES: SELF-PREDICTION METHODS

Another line of model pre-training research is based on the recent progress of self-prediction methods in natural language processing (Devlin et al., 2018; Brown et al., 2020). Along that line, given a protein, our objective can be formulated as predicting one part of the protein given the remainder of the structure. Here, we propose four self-supervised tasks based on physicochemical or geometric properties: residue types, distances, angles and dihedrals.

The four methods perform masked prediction on single residues, residue pairs, triplets and quadruples, respectively. Masked residue type prediction is widely used for pretraining protein language

models (Bepler & Berger, 2021) and also known as masked inverse folding in the protein community (Yang et al., 2022). Distances, angles and dihedrals have been shown to be important features that reflect the relative position between residues (Klicpera et al., 2020). For angle and dihedral prediction, we sample adjacent edges to better capture local structual information. Since angular values are more sensitive to errors in protein structures than distances, we use discretized values for prediction. The four objectives are summarized in Table 1 and details will be discussed in Appendix. D.

| Method | Loss function | Sampled items |
|---|---|---|
| Residue Type Prediction | $\mathcal{L}_i = \mathrm{CE}(f_{\mathrm{residue}}(\boldsymbol{h}'_i), \boldsymbol{f}_i)$ | Single residue |
| Distance Prediction | $\mathcal{L}_{(i,j,r)} = (f_{\mathrm{dist}}(\boldsymbol{h}'_i, \boldsymbol{h}'_j) - \|\boldsymbol{x}_i - \boldsymbol{x}_j\|_2)^2$ | Single edge |
| Angle Prediction | $\mathcal{L}_{(i,j,r_1),(j,k,r_2)} = \mathrm{CE}(f_{\mathrm{angle}}(\boldsymbol{h}'_i, \boldsymbol{h}'_j, \boldsymbol{h}'_k), \mathrm{bin}(\angle ijk))$ | Adjacent edge pairs |
| Dihedral Prediction | $\mathcal{L}_{(i,j,r_1),(j,k,r_2),(k,t,r_3)} = \mathrm{CE}(f_{\mathrm{dih}}(\boldsymbol{h}'_i, \boldsymbol{h}'_j, \boldsymbol{h}'_k, \boldsymbol{h}'_t), \mathrm{bin}(\angle ijkt))$ | Adjacent edge triplets |

Table 1: Self-prediction methods. We use $f_{\mathrm{residue}}(\cdot), f_{\mathrm{dist}}(\cdot), f_{\mathrm{angle}}(\cdot), f_{\mathrm{dih}}(\cdot)$ to denote the MLP head in each task. $\mathrm{CE}(\cdot)$ denotes the cross entropy loss and $\mathrm{bin}(\cdot)$ is used to discretize the angle. $\boldsymbol{f}_i$ and $\boldsymbol{x}_i$ denote the residue type feature and coordinate of node $i$, respectively. $\boldsymbol{h}'_i$ denotes the representation of node $i$ after masking the corresponding sampled items in each task.

## 5 EXPERIMENTS

In this section, we first introduce our experimental setup for pretraining and then evaluate our models on four standard downstream tasks including Enzyme Commission number prediction, Gene Ontology term prediction, fold and reaction classification. More analysis can be found in Appendix F-K.

### 5.1 EXPERIMENTAL SETUP

**Pretraining datasets.** We use the AlphaFold protein structure database (CC-BY 4.0 License) (Varadi et al., 2021) for pretraining. This database contains protein structures predicted by AlphaFold2, and we employ both 365K proteome-wide predictions and 440K Swiss-Prot (Consortium, 2021) predictions. In Appendix F, we further report the results of pretraining on different datasets.

**Downstream tasks.** We adopt two tasks proposed in Gligorijević et al. (2021) and two tasks used in Hermosilla et al. (2021) for downstream evaluation. *Enzyme Commission (EC) number prediction* seeks to predict the EC numbers of different proteins, which describe their catalysis of biochemical reactions. The EC numbers are selected from the third and fourth levels of the EC tree (Webb et al., 1992), forming 538 binary classification tasks. *Gene Ontology (GO) term prediction* aims to predict whether a protein belongs to some GO terms. These terms classify proteins into hierarchically related functional classes organized into three ontologies: molecular function (MF), biological process (BP) and cellular component (CC). *Fold classification* is first proposed in Hou et al. (2018), with the goal to predict the fold class label given a protein. *Reaction classification* aims to predict the enzyme-catalyzed reaction class of a protein, in which all four levels of the EC number are employed to depict the reaction class. Although this task is essentially the same with EC prediction, we include it to make a fair comparison with the baselines in Hermosilla et al. (2021).

**Dataset splits.** For EC and GO prediction, we follow the multi-cutoff split methods in Gligorijević et al. (2021) to ensure that the test set only contains PDB chains with sequence identity no more than 95% to the training set as used in Wang et al. (2022b) (See Appendix. F for results at lower identity cutoffs). For fold classification, Hou et al. (2018) provides three different test sets: *Fold*, in which proteins from the same superfamily are unseen during training; *Superfamily*, in which proteins from the same family are not present during training; and *Family*, in which proteins from the same family are present during training. For reaction classification, we adopt dataset splits proposed in Hermosilla et al. (2021), where proteins have less than 50% sequence similarity in-between splits.

**Baselines.** Following Wang et al. (2022b) and Hermosilla et al. (2021), we compare our encoders with many existing protein representation learning methods, including four sequence-based encoders (CNN (Shanehsazzadeh et al., 2020), ResNet (Rao et al., 2019), LSTM (Rao et al., 2019) and Transformer (Rao et al., 2019)), six structure-based encoders (GCN (Kipf & Welling, 2017), GAT (Veličković et al., 2018), GVP (Jing et al., 2021), 3DCNN_MQA (Derevyanko et al., 2018), GraphQA (Baldassarre et al., 2021) and New IEConv (Hermosilla & Ropinski, 2022)). We also include two models pretrained on large-scale sequence datasets (ProtBERT-BFD (Elnaggar et al.,

Table 2: $F_{max}$ on EC and GO prediction and Accuracy (%) on fold and reaction classification. [†] denotes results taken from Wang et al. (2022b) and [*] denotes results taken from Hermosilla et al. (2021) and Hermosilla & Ropinski (2022). Bold numbers indicate the best results under w/o pretraining and w/ pretraining settings. For pretraining, we select the model with the best performance when training from scratch, *i.e.*, GearNet-Edge for EC, GO, Reaction and GearNet-Edge-IEConv for Fold Classification. We use the pretraining methods to name our pretrained models.

| | Method | Pretraining Dataset (Size) | EC | GO | | | Fold Classification | | | | Reaction |
| | | | | BP | MF | CC | Fold | Super. | Fam. | Avg. | |
|---|---|---|---|---|---|---|---|---|---|---|---|
| w/o pretraining | CNN (Shanehsazzadeh et al., 2020) | - | 0.545 | 0.244 | 0.354 | 0.287 | 11.3 | 13.4 | 53.4 | 26.0 | 51.7 |
| | ResNet (Rao et al., 2019) | - | 0.605 | 0.280 | 0.405 | 0.304 | 10.1 | 7.21 | 23.5 | 13.6 | 24.1 |
| | LSTM (Rao et al., 2019) | - | 0.425 | 0.225 | 0.321 | 0.283 | 6.41 | 4.33 | 18.1 | 9.61 | 11.0 |
| | Transformer (Rao et al., 2019) | - | 0.238 | 0.264 | 0.211 | 0.405 | 9.22 | 8.81 | 40.4 | 19.4 | 26.6 |
| | GCN (Kipf & Welling, 2017) | - | 0.320 | 0.252 | 0.195 | 0.329 | 16.8* | 21.3* | 82.8* | 40.3* | 67.3* |
| | GAT (Veličković et al., 2018) | - | 0.368 | 0.284† | 0.317† | 0.385† | 12.4 | 16.5 | 72.7 | 33.8 | 55.6 |
| | GVP (Jing et al., 2021) | - | 0.489 | 0.326† | 0.426† | 0.420† | 16.0 | 22.5 | 83.8 | 40.7 | 65.5 |
| | 3DCNN_MQA (Derevyanko et al., 2018) | - | 0.077 | 0.240 | 0.147 | 0.305 | 31.6* | 45.4* | 92.5* | 56.5* | 72.2* |
| | GraphQA (Baldassarre et al., 2021) | - | 0.509 | 0.308 | 0.329 | 0.413 | 23.7* | 32.5* | 84.4* | 46.9* | 60.8* |
| | New IEConv (Hermosilla & Ropinski, 2022) | - | 0.735 | 0.374 | 0.544 | 0.444 | 47.6* | 70.2* | 99.2* | 72.3* | **87.2*** |
| | **GearNet** | - | 0.730 | 0.356 | 0.503 | 0.414 | 28.4 | 42.6 | 95.3 | 55.4 | 79.4 |
| | **GearNet-IEConv** | - | 0.800 | 0.381 | 0.563 | 0.422 | 42.3 | 64.1 | 99.1 | 68.5 | 83.7 |
| | **GearNet-Edge** | - | **0.810** | **0.403** | **0.580** | **0.450** | 44.0 | 66.7 | 99.1 | 69.9 | 86.6 |
| | **GearNet-Edge-IEConv** | - | **0.810** | **0.400** | **0.581** | 0.430 | **48.3** | **70.3** | **99.5** | **72.7** | 85.3 |
| w/ pretraining | DeepFRI (Gligorijević et al., 2021) | Pfam (10M) | 0.631 | 0.399 | 0.465 | 0.460 | 15.3* | 20.6* | 73.2* | 36.4* | 63.3* |
| | ESM-1b (Rives et al., 2021) | UniRef50 (24M) | 0.864 | 0.452 | **0.657** | 0.477 | 26.8 | 60.1 | 97.8 | 61.5 | 83.1 |
| | ProtBERT-BFD (Elnaggar et al., 2021) | BFD (2.1B) | 0.838 | 0.279† | 0.456† | 0.408† | 26.6* | 55.8* | 97.6* | 60.0* | 72.2* |
| | LM-GVP (Wang et al., 2022b) | UniRef100 (216M) | 0.664 | 0.417† | 0.545† | **0.527†** | - | - | - | - | - |
| | New IEConv (Hermosilla & Ropinski, 2022) | PDB (476K) | - | - | - | - | 50.3* | **80.6*** | 99.7* | 76.9* | **87.6*** |
| | Residue Type Prediction | AlphaFoldDB (805K) | 0.843 | 0.430 | 0.604 | 0.465 | 48.8 | 71.0 | 99.4 | 73.0 | 86.6 |
| | Distance Prediction | AlphaFoldDB (805K) | 0.839 | 0.448 | 0.616 | 0.464 | 50.9 | 73.5 | 99.4 | 74.6 | 87.5 |
| | Angle Prediction | AlphaFoldDB (805K) | 0.853 | 0.458 | 0.625 | 0.473 | **56.5** | 76.3 | 99.6 | 77.4 | 86.8 |
| | Dihedral Prediction | AlphaFoldDB (805K) | 0.859 | 0.458 | 0.626 | 0.465 | 51.8 | 77.8 | 99.6 | 75.9 | 87.0 |
| | **Multiview Contrast** | AlphaFoldDB (805K) | **0.874** | **0.490** | 0.654 | 0.488 | 54.1 | 80.5 | 99.9 | 78.1 | 87.5 |

2021), ESM-1b (Rives et al., 2021)) and two models combining pretrained sequence-based encoders with structural information (DeepFRI (Gligorijević et al., 2021) and LM-GVP (Wang et al., 2022b)). For LM-GVP and New IEConv, we only include results reported in the original paper due to the computational burden and the lack of codes. We do not include MSA-based baselines, since these evolution-based methods require a lot of resources for the computation and storage of MSAs but have been shown to be inferior to ESM-1b on function prediction tasks in Hu et al. (2022).

**Training.** On the four downstream tasks, we train GearNet and GearNet-Edge from scratch. As we find that the IEConv layer is important for predicting fold labels, we also enhance our model by incorporating this as an additional layer (see Appendix C.2). These models are referred as GearNet-IEConv and GearNet-Edge-IEConv, respectively. Following Wang et al. (2022b) and Hermosilla & Ropinski (2022), the models are trained for 200 epochs on EC and GO prediction and for 300 epochs on fold and reaction classification. For pretraining, the models with the best performance when trained from scratch are selected, *i.e.*, GearNet-Edge for EC, GO, Reaction and GearNet-Edge-IEConv for Fold Classification. The models are pretrained on the AlphaFold database with our proposed five methods for 50 epochs. All these models are trained on 4 Tesla A100 GPUs (see Appendix E.3).

**Evaluation.** For EC and GO prediction, we evaluate the performance with the protein-centric maximum F-score $F_{max}$, which is commonly used in the CAFA challenges (Radivojac et al., 2013) (See Appendix E.2 for details). For fold and reaction classification, the performance is measured with the mean accuracy. Models with the best performance on validation sets are selected for evaluation.

## 5.2 RESULTS

We report results for four downstream tasks in Table 2, including all models with and without pretraining. The following conclusions can be drawn from the results:

**Our structure-based encoders outperform all baselines without pretraining on 7 of 8 datasets.** By comparing the first three blocks, we find that GearNet can obtain competitive results against other baselines on three function prediction tasks (EC, GO, Reaction). After adding the edge message passing mechanism, GearNet-Edge significantly outperforms other baselines on EC, GO-BP and GO-MF and is competitive on GO-CC. Although no clear improvements are observed on function prediction (EC, GO, Reaction) by adding IEConv layers, GearNet-Edge-IEConv achieve the best results on fold classification. This can be understood since fold classification requires the encoder to capture sufficient structural information for determining the fold labels. Compared with GearNet-

Edge, which only includes the distance and angle information as features, the IEConv layer is better at capturing structural details by applying different kernel matrices dependent on relative positional features. These strong performance demonstrates the advantages of our structure-based encoders.

**Structure-based encoders benefit a lot from pretraining with unlabeled structures.** Comparing the results in the third and last two blocks, it can be observed that models with all proposed pretraining methods show large improvements over models trained from scratch. Among these methods, Multiview Contrast is the best on 7 of 8 datasets and achieve the state-of-the-art results on EC, GO-BP, GO-MF, Fold and Reaction tasks. This proves the effectiveness of our pretraining strategies.

**Pretrained structure-based encoders perform on par with or even better than sequence-based encoders pretrained with much more data.** The last three blocks show the comparision between pretrained sequence-based and structure-based models. It should be noted that our models are pretrained on a dataset with fewer than one million structures, whereas all sequence-based pretraining baselines are pretrained on million- or billion-scale sequence databases. Though pretrained with an order of magnitude less data, our model can achieve comparable or even better results against these sequence-based models. Besides, our model is the only one that can achieve good performance on all four tasks, given that sequence-based models do not perform well on fold classification. This again shows the potential of structure-based pretraining for learning protein representations.

## 5.3 Ablation Studies

| Method | # layers | # params. | EC |
|---|---|---|---|
| **GearNet-Edge** | 6 | 42M | **0.810** |
| - w/o rel. conv. | 6 | 23M | 0.752 |
| - w/o rel. conv. | 8 | 39M | 0.754 |
| - w/o rel. conv. | 10 | 60M | 0.744 |

| Method | EC | GO-BP | GO-MF | GO-CC |
|---|---|---|---|---|
| **Multiview Contrast** | **0.874** | **0.490** | **0.654** | **0.488** |
| - subsequence + identity | 0.866 | 0.477 | 0.627 | 0.473 |
| - subspace+ identity | **0.872** | 0.480 | 0.640 | 0.468 |
| - subsequence + random edge masking | 0.869 | 0.484 | 0.641 | 0.471 |
| - subspace + random edge masking | **0.876** | 0.481 | 0.645 | 0.470 |

Table 3: Ablation studies of GearNet-Edge and Multiview Contrast. Rel. conv. is the abbreviation for relational graph convolution. The numbers of layers and parameters in GearNet-Edge are reported.

To analyze the contribution of different components in our proposed methods, we perform ablation studies on function prediction task. The results are shown in Table 3.

**Relational graph convolutional layers.** To show the effects of relational convolutional layers, we replace it with graph convolutional layers that share a single kernel matrix among all edges. To make the number of learnable parameters comparable, we run the baselines with different number of layers. As reported in the table, results can be significantly improved by using relational convolution, which suggests the importance of treating edges as different types.

**Edge message passing layers.** We also compare the results of GearNet with and without edge message passing layers, the results of which are shown in Table 2. It can be observed that the performance consistently increases after performing edge message passing. This demonstrates the effectiveness of our proposed mechanism.

**Different augmentations in Multiview Contrast.** We investigate the contribution of each augmentation operation proposed in the Multiview Contrast method. Instead of randomly sampling cropping and noise functions, we pretrain our model with four deterministic combinations of augmentations, respectively. As shown in Table 3, all the four combinations can yield good results, which suggests that arbitrary combinations of the proposed cropping and noise schemes can yield informative partial views of proteins. Besides, by randomly choosing cropping and noise functions, we can generate more diverse views and thus benefit contrastive learning, as demonstrated in the table.

## 6 Conclusions and Future Work

In this work, we propose a simple yet effective structure-based encoder for protein representation learning, which performs relational message passing on protein residue graphs. A novel edge message passing mechanism is introduced to explicitly model interactions between edges, which show consistent improvements. Moreover, five self-supervised pretraining methods are proposed following two standard frameworks: contrastive learning and self-prediction methods. Comprehensive experiments over multiple benchmark tasks verify that our model outperforms previous encoders when trained from scratch and achieve comparable or even better results than the state-of-the-art baselines while pretraining with much less data. We believe that our work is an important step towards adopting self-supervised learning methods on protein structure understanding.

ACKNOWLEDGMENTS

The authors would like to thank Meng Qu, Zhaocheng Zhu, Shengchao Liu, Chence Shi, Minkai Xu and Huiyu Cai for their helpful discussions and comments.

This project is supported by AIHN IBM-MILA partnership program, the Natural Sciences and Engineering Research Council (NSERC) Discovery Grant, the Canada CIFAR AI Chair Program, collaboration grants between Microsoft Research and Mila, Samsung Electronics Co., Ltd., Amazon Faculty Research Award, Tencent AI Lab Rhino-Bird Gift Fund, a NRC Collaborative R&D Project (AI4D-CORE-06) as well as the IVADO Fundamental Research Project grant PRF-2019-3583139727.

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

# A    MORE RELATED WORK

## A.1    SEQUENCE-BASED METHODS FOR PROTEIN REPRESENTATION LEARNING

Sequence-based protein representation learning is mainly inspired by the methods of modeling natural language sequences. Recent methods aim to capture the biochemical and co-evolutionary knowledge underlying a large-scale protein sequence corpus by self-supervised pretraining, and such knowledge is then transferred to specific downstream tasks by finetuning. Typical pretraining objectives explored in existing methods include next amino acid prediction (Alley et al., 2019; Elnaggar et al., 2021), masked language modeling (MLM) (Rao et al., 2019; Elnaggar et al., 2021; Rives et al., 2021), pairwise MLM (He et al., 2021) and contrastive predictive coding (CPC) (Lu et al., 2020). Compared to sequence-based approaches that learn in the whole sequence space, MSA-based methods (Rao et al., 2021; Biswas et al., 2021; Meier et al., 2021) leverage the sequences within a protein family to capture the conserved and variable regions of homologous sequences, which imply specific structures and functions of the protein family.

## A.2    STRUCTURE-BASED METHODS FOR BIOLOGICAL MOLECULES

Following the early efforts (Behler & Parrinello, 2007; Bartók et al., 2010; 2013; Chmiela et al., 2017) of building machine learning systems for molecules by hand-crafted atomic features, recent works exploited end-to-end message passing neural networks (MPNNs) (Gilmer et al., 2017) to encode the structures of small molecules and macromolecules like proteins. Specifically, existing methods employed node/atom message passing (Gilmer et al., 2017; Schütt et al., 2017a;b), edge/bond message passing (Jørgensen et al., 2018; Chen et al., 2019) and directional information (Klicpera et al., 2020; Liu et al., 2021; Klicpera et al., 2021) to encode 2D or 3D molecular graphs.

Compared to small molecules, structural representations of proteins are more diverse, including residue-level, atom-level graphs and protein surfaces. There are recent models designed for residue-level graphs (Hermosilla et al., 2021; Hermosilla & Ropinski, 2022) and protein surfaces (Gainza et al., 2020; Sverrisson et al., 2021), which achieved impressive results on various tasks.

## A.3    PRETRAINING GRAPH NEURAL NETWORKS

Our work is also related to the recent efforts of pretraining graph neural networks (GNNs), which sought to learn graph representations in a self-supervised fashion. In this domain, various self-supervised pretext tasks, like edge prediction (Kipf & Welling, 2016; Hamilton et al., 2017), context prediction (Hu et al., 2019; Rong et al., 2020), node/edge attribute reconstruction (Hu et al., 2019) and contrastive learning (Hassani & Khasahmadi, 2020; Qiu et al., 2020; You et al., 2020; Xu et al., 2021), are designed to acquire knowledge from unlabeled graphs. Besides, there is one recent work applying self-supervised learning for protein-ligand affinity prediction (You & Shen, 2022). In this work, we focus on learning representations of residue-level graphs of proteins in a self-supervised way. To attain this goal, we design novel protein-specific pretraining methods to learn the proposed structure-based encoder.

# B    POTENTIAL NEGATIVE IMPACT

This research project focuses on learning effective protein representations via pretraining with a large number of unlabeled protein structures. Compared to the conventional sequence-based pretraining methods, our approach is able to leverage structural information and thus provide better representations. This merit enables more in-depth analysis of protein research and can potentially benefit many real-world applications, like protein function prediction and sequence design.

**Limitations.**    In this paper, we only use 805K protein structures for pretraining. As the AlphaFold Protein Structure Database now covers over 100 million proteins, it would be possible to train huge and more advanced structure-based models on larger datasets in the future. Scalability, however, should be kept in mind. Thanks to their simplicity, our models could readily accommodate larger datasets, while more cumbersome procedures might not. Besides, we only consider function and fold prediction tasks. Another promising direction is to apply our proposed methods on more tasks, *e.g.*,

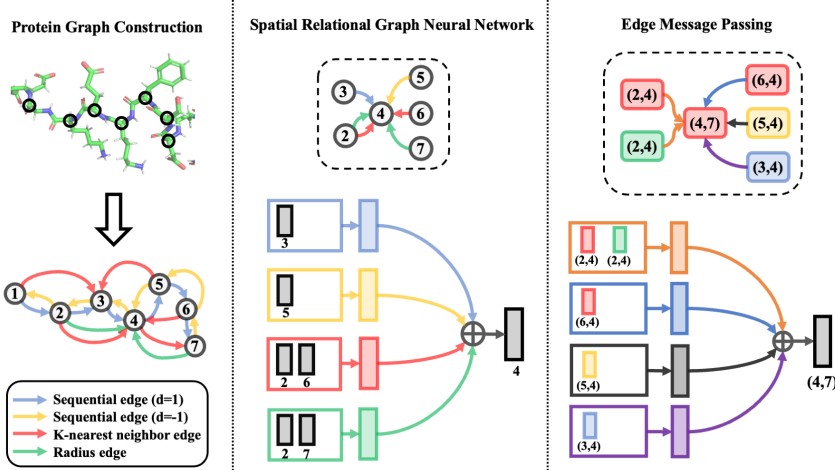

Figure 3: The pipeline for GearNet and GearNet-edge. First, we construct a relational protein residue graph with sequential, radius and knn edges (some edges are omitted in the figure to save space). Then, a relational graph convolutional layer is applied. Similar message passing layers can be applied on the edge graph to improve the model capacity. This figure shows the update iteration for node 4 and edge $(4, 7, \text{red})$, respectively.

protein-protein interaction modeling and protein-guided ligand molecule design, which underpins many important biological processes and applications.

It cannot be denied that some harmful activities could be augmented by powerful pretrained models, *e.g.*, designing harmful drugs. We expect future studies will mitigate these issues.

## C    MORE DETAILS OF GEARNET

In this section, we describe more details about the implementation of our GearNet. The whole pipeline of our structure-based encoder is depicted in Figure 3.

### C.1    PROTEIN GRAPH CONSTRUCTION

For graph construction, we use three different ways to add edges:

1. **Sequential edges.** The $i$-th residue and the $j$-th residue will be linked by an edge if the sequential distance between them is below a predefined threshold $d_{\text{seq}}$, i.e., $|j - i| < d_{\text{seq}}$. The type of each sequential edge is determined by their relative position $d = j - i$ in the sequence. Hence, there are $2d_{\text{seq}} - 1$ types of sequential edges.
2. **Radius edges.** Following previous works, we also add edges between two nodes $i$ and $j$ when the Euclidean distance between them is smaller than a threshold $d_{\text{radius}}$.
3. **K-nearest neighbor edges.** Since the scales of spatial coordinates may vary among different proteins, a node will be also connected to its k-nearest neighbors based on the Euclidean distance. In this way, the density of spatial edges are guaranteed to be comparable among different protein graphs.

Since we are not interested in spatial edges between residues close with each other in the sequence, we further add a filter to the latter two kinds of edges. Specifically, for a radius or KNN edge connecting the $i$-th residue and $j$-th residue, it will be removed if the sequential distance between them is lower than a long range interaction cutoff $d_{\text{long}}$, *i.e.*, $|i - j| < d_{\text{long}}$.

In this paper, we set the sequential distance threshold $d_{\text{seq}} = 3$, the radius $d_{\text{radius}} = 10.0$, the number of neighbors $k = 10$ and the long range interaction cutoff $d_{\text{long}} = 5$. By regarding radius edges and KNN edges as two separate edge types, there will be totally $2d_{\text{seq}} + 1 = 7$ different types of edges.

**Necessity of spatial edges.** Here we explain the necessity of radius and KNN edges by statistics and intuitions. These two kinds of edges result in very different degree distributions. In Figure 4, we plot the average degree distribution over all proteins in AlphaFold Database v1. If we only consider KNN edges, the node degrees in protein graphs are close to a constant, which makes it difficult to capture those areas with dense interactions between residues. If we only consider radius edges, then there will be about 45,000 proteins with average degrees lower than two. In these sparse graphs, pretraining cannot capture structural information effectively, *e.g.*, Angle Prediction with limited edge pairs and Dihedral Prediction with limited

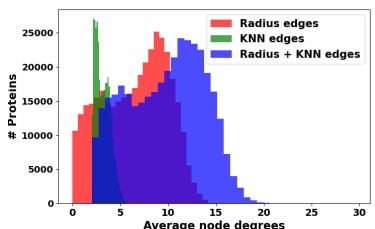

Figure 4: The average degree distribution on AlphaFold Database.

edge triplets. Such sparsity can be hard to overcome by tuning radius cutoff, for the various scales of average distance on different proteins. By simply combining two kinds of edges, we can overcome these issues.

**Node and edge features.** Most previous structure-based encoders designed for biological molecules (Baldassarre et al., 2021; Hermosilla et al., 2021) used many chemical and spatial features, some of which are difficult to obtain or time-consuming to calculate. In contrast, we only use the one-hot encoding of residue types with one additional dimension for unknown types as node features, denoted as $\boldsymbol{f} \in \{0, 1\}^{n \times 21}$, which is enough to learn good representation as shown in our experiments.

The feature $\boldsymbol{f}_{(i,j,r)}$ for an edge $(i, j, r)$ is the concatenation of the node features of two end nodes, the one-hot encoding of the edge type, and the sequential and spatial distances between them:

$$\boldsymbol{f}_{(i,j,r)} = \text{Cat}\left(\boldsymbol{f}_i, \boldsymbol{f}_j, \text{onehot}(r), |i - j|, \|\boldsymbol{x}_i - \boldsymbol{x}_j\|_2\right), \quad (4)$$

where $\text{Cat}(\cdot)$ denotes the concatenation operation.

## C.2 ENHANCE GEARNET WITH IECONV LAYERS

In our experiments, we find that IEConv layers are useful for predicting fold labels in spite of their relatively poor performance on function prediction tasks. Therefore, we enhance our models by adding a simplified IEConv layer as an additional layer, which achieve better results than the original IEConv. Next, we describe how to simplify the IEConv layer and how to combine it with our model.

**Simplify the IEConv layer.** The original IEConv layer relies on intrinsic and extrinsic distances between two nodes, which are computationally expensive. Hence, we follow the modifications proposed in Hermosilla & Ropinski (2022), which show improvements as reported in their experiments. We briefly describe these modifications for completeness.

In the IEConv layer, we keep the edges in our graph $\mathcal{G}$ and use $\tilde{\boldsymbol{h}}_i^{(l)}$ to denote the hidden representation for node $i$ in the $l$-th layer. The update equation for node $i$ is defined as:

$$\tilde{\boldsymbol{h}}_i^{(l)} = \sum_{j \in \mathcal{N}(i)} k_o(f(\mathcal{G}, i, j)) \cdot \boldsymbol{h}_j^{(l-1)}, \quad (5)$$

where $\mathcal{N}(i)$ is the set of neighbors of $i$, $f(\mathcal{G}, i, j)$ is the edge feature between $i$ and $j$ and $k_o(\cdot)$ is an MLP mapping the feature to a kernel matrix. Instead of intrinsic and extrinsic distances in the original IEConv layer, we follow New IEConv, which adopts three relative positional features proposed in Ingraham et al. (2019) and further augments them with additional input functions.

We aim to apply this layer on our constructed protein residue graph instead of the radius graph in the original paper. Therefore, we simply remove the dynamically changed receptive fields, pooling layer and smoothing tricks in our setting.

**Combine IEConv with GearNet.** Our model is very flexible to incorporate other message passing layers. To incorporate IEConv layers, we use our graph and hidden representations as input and replace the update equation Eq. (1) with

$$\boldsymbol{h}_i^{(l)} = \boldsymbol{h}_i^{(l-1)} + \boldsymbol{u}_i^{(l)} + \tilde{\boldsymbol{h}}_i^{(l)}. \quad (6)$$

# D SELF-PREDICTION METHODS

The high-level ideas of the four self-prediction methods are demonstrated in Figure 5.

**Residue Type Prediction** is based on the masked language modeling objective, which has been widely used in pretraining large protein language models (Bepler & Berger, 2021). For each protein, we randomly mask node features of some residues and then predict these masked residue types via structure-based encoders. This method is also known as Attribute Masking in the literature of molecules (Hu et al., 2019) and Masked Inverse Folding in the protein community (Yang et al., 2022).

**Distance Prediction** aims to learn local spatial structures by predicting the Euclidean distance between two nodes connected in the protein graph. A fixed number of edges are randomly selected and masked from the original graph. Then, the representations of two end nodes will be used to predict the distances between them.

Besides, angles and dihedrals between adjacent edges are also important features that reflect the relative position between residues Klicpera et al. (2021). Similarly, we can define the masked geometric losses **Angle Prediction** and **Dihedral Prediction** by randomly selecting and masking adjacent edge pairs and triplets. Here we discretize the angles and dihedrals by cutting the range $[0, \pi]$ into 8 bins. Then, the representations of end nodes in the masked graph will be used to predict which bin the angles between them will belong to.

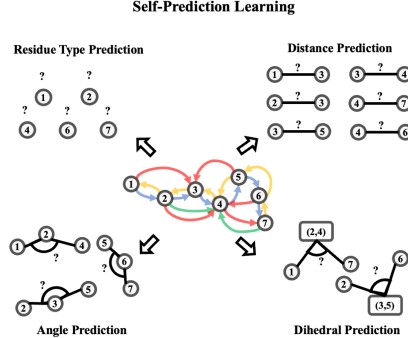

Figure 5: Illustration of self-prediction methods.

# E EXPERIMENTAL DETAILS

## E.1 DATASET STATISTICS

Table 4: Dataset statistics for downstream tasks.

| Dataset | # Proteins | | |
| --- | --- | --- | --- |
| | # Train | # Validation | # Test |
| **Enzyme Commission** | 15,550 | 1,729 | 1,919 |
| **Gene Ontology** | 29,898 | 3,322 | 3,415 |
| **Fold Classification -** *Fold* | 12,312 | 736 | 718 |
| **Fold Classification -** *Superfamily* | 12,312 | 736 | 1,254 |
| **Fold Classification -** *Family* | 12,312 | 736 | 1,272 |
| **Reaction Classification** | 29,215 | 2,562 | 5,651 |

Dataset statistics of downstream tasks are summarized in Table 4. Details are introduced as follows.

**Enzyme Commission and Gene Ontology.** Following DeepFRI (Gligorijević et al., 2021), the EC numbers are selected from the third and fourth levels of the EC tree, forming 538 binary classification tasks, while the GO terms with at least 50 and no more than 5000 training samples are selected. The non-redundant sets are partitioned into training, validation and test sets according to the sequence identity. We retrieve all protein chains from PDB with the code in their codebase and remove those with obsolete pdb ids, so the statistics will be slightly different from that in the original paper.

**Fold Classification.** We directly use the dataset in Hermosilla et al. (2021), which consolidated 16,712 proteins with 1,195 different folds from the SCOPe 1.75 database (Murzin et al., 1995).

**Reaction Classification.** The dataset comprises 37,428 proteins categorized into 384 reaction classes. The split methods are described in Hermosilla et al. (2021), where they cluster protein chains via sequence similarities and ensure that protein chains from the same cluster are in the same split.

### E.2 EVALUATION METRICS

Now we introduce the details of evaluation metrics for EC and GO prediction. These two tasks aim to answer the question: whether a protein has some particular functions, which can be seen as multiple binary classification tasks.

The first metric, protein-centric maximum F-score $F_{max}$, is defined by first calculating the precision and recall for each protein and then taking the average score over all proteins. More specifically, for a given target protein $i$ and a decision threshold $t \in [0, 1]$, the precision and recall are computed as:

$$\text{precision}_i(t) = \frac{\sum_f \mathbb{1}[f \in P_i(t) \cap T_i]}{\sum_f \mathbb{1}[f \in P_i(t)]}, \tag{7}$$

and

$$\text{recall}_i(t) = \frac{\sum_f \mathbb{1}[f \in P_i(t) \cap T_i]}{\sum_f \mathbb{1}[f \in T_i]}, \tag{8}$$

where $f$ is a function term in the ontology, $T_i$ is a set of experimentally determined function terms for protein $i$, $P_i(t)$ denotes the set of predicted terms for protein $i$ with scores greater than or equal to $t$ and $\mathbb{1}[\cdot] \in \{0, 1\}$ is an indicator function that is equal to 1 *iff* the condition is true.

Then, the average precision and recall over all proteins at threshold $t$ is defined as:

$$\text{precision}(t) = \frac{1}{M(t)} \sum_i \text{precision}_i(t), \tag{9}$$

and

$$\text{recall}(t) = \frac{1}{N} \sum_i \text{recall}_i(t), \tag{10}$$

where we use $N$ to denote the number of proteins and $M(t)$ to denote the number of proteins on which at least one prediction was made above threshold $t$, *i.e.*, $|P_i(t)| > 0$.

Combining these two measures, the maximum F-score is defined as the maximum value of F-measure over all thresholds. That is,

$$F_{max} = \max_t \left\{ \frac{2 \cdot \text{precision}(t) \cdot \text{recall}(t)}{\text{precision}(t) + \text{recall}(t)} \right\}. \tag{11}$$

The second metric, pair-centric area under precision-recall curve $AUPR_{pair}$, is defined as the average precision scores for all protein-function pairs, which is exactly the micro average precision score for multiple binary classification.

### E.3 IMPLEMENTATION DETAILS

In this subsection, we describe implementation details of all baselines and our methods. For all models, the outputs will be fed into a three-layer MLP to make final prediction. The dimension of hidden layers in the MLP is equal to the dimension of model outputs.

**CNN (Shanehsazzadeh et al., 2020).** Following the finding in Shanehsazzadeh et al. (2020), we employ a convolutional neural network (CNN) to encode protein sequences. Specifically, 2 convolutional layers with 1024 hidden dimensions and kernel size 5 constitute this baseline model.

**ResNet (Rao et al., 2019).** We also adopt a deep CNN model, *i.e.*, the ResNet for protein sequences proposed by Rao et al. (2019), in our benchmark. This model is with 12 residual blocks and 512 hidden dimensions, and it uses the GELU (Hendrycks & Gimpel, 2016) activation function.

**LSTM (Rao et al., 2019).** The bidirectional LSTM model proposed by Rao et al. (2019) is another baseline for protein sequence encoding. It is composed of three bidirectional LSTM layers with 640 hidden dimensions.

**Transformer (Rao et al., 2019).** The self-attention-based Transformer encoder (Vaswani et al., 2017) is a strong model in natural language processing (NLP), Rao et al. (2019) adapts this model into the field of protein sequence modeling. We also adopt it as one of our baselines. This model has a comparable size with BERT-Small (Devlin et al., 2018), which contains 4 Transformer blocks with 512 hidden dimensions and 8 attention heads, with activation GELU (Hendrycks & Gimpel, 2016).

**GCN (Kipf & Welling, 2017).** We take GCN as a baseline to encode the residue graph derived by our graph construction scheme. We adopt the implementation in TorchDrug (Zhu et al., 2022), where 6 GCN layers with the hidden dimension of 512 are used. We run the results of GCN on EC and GO by ourselves and take its results on Fold and Reaction classification from Hermosilla et al. (2021).

**GAT (Veličković et al., 2018).** We adopt another popular graph neural network, GAT, as a structure-based baseline. We follow the implementation in TorchDrug and use 6 GAT layers with the hidden dimension of 512 and 1 attention head per layer for encoding. The results on EC, Fold and Reaction classification are based on our runs, and the results on GO are taken from Wang et al. (2022b).

**GVP (Jing et al., 2021).** The GVP model (Jing et al., 2021) is a decent protein structure encoder. It iteratively updates the scalar and vector representations of a protein, and these representations possess the merit of invariance and equivariance. In our benchmark, we evaluate this baseline method following the official source code. In specific, 3 GVP layers with 32 feature dimensions (20 scalar and 4 vector channels) constitute the GVP model.

**3DCNN_MQA (Derevyanko et al., 2018).** We implement the 3DCNN model from the paper (Derevyanko et al., 2018) with a box width of 40.0 and input resolution of $120 \times 120 \times 120$. The model has 6 residual blocks and 128 hidden dimensions with ELU activation function.

**GraphQA (Baldassarre et al., 2021).** Following hyperparameters in the original paper, we construct residue graphs based on bond and spatial information and re-implement the graph neural network. The best model has 4 layers with 128 node features, 32 edge features and 512 global features.

**New IEConv (Hermosilla & Ropinski, 2022).** Since the code for New IEConv has not been made public when the paper is written, we reproduce the method according to the description in the paper and achieve similar results on Fold and Reaction classification tasks. Then, we evaluate the method on EC and GO prediction tasks with the default hyperparameters reported in the original paper and follow the standard training procedure on these two tasks.

**DeepFRI (Gligorijević et al., 2021).** We also evaluate DeepFRI (Gligorijević et al., 2021) in our benchmark, which is a popular structure-based encoder for protein function prediction. DeepFRI employs an LSTM model to extract residue features and further constructs a residue graph to propagate messages among residues, in which a 3-layer graph convolutional network (GCN) (Kipf & Welling, 2017) is used. We directly utilize the official model checkpoint for baseline evaluation.

**ESM-1b (Rives et al., 2021).** Besides the from-scratch sequence encoders above, we also compare with two state-of-the-art pretrained protein language models. ESM-1b (Rives et al., 2021) is a huge Transformer encoder model whose size is larger than BERT-Large (Devlin et al., 2018), and it is pretrained on 24 million protein sequences from UniRef50 (Suzek et al., 2007) by masked language modeling (MLM) (Devlin et al., 2018). In our evaluation, we finetune the ESM-1b model with the learning rate that is one-tenth of that of the MLP prediction head.

**ProtBERT-BFD (Elnaggar et al., 2021).** The other protein language model evaluated in our benchmark is ProtBERT-BFD (Elnaggar et al., 2021) whose size also excesses BERT-Large (Devlin et al., 2018). This model is pretrained on 2.1 billion protein sequences from BFD (Steinegger & Söding, 2018) by MLM (Devlin et al., 2018). The evaluation of ProtBERT-BFD uses the same learning rate configuration as ESM-1b.

**LM-GVP (Wang et al., 2022b).** To further enhance the effectiveness of GVP (Jing et al., 2021), Wang et al. (2022b) proposed to prepend a protein language model, *i.e.* ProtBERT (Elnaggar et al.,

Table 5: Hyperparameter configurations of our model on different datasets. The batch size reported in the table refers to the batch size on each GPU. All the hyperparameters are chosen by the performance on the validation set.

| Hyperparameter | | EC | GO | Fold | Reaction |
|---|---|---|---|---|---|
| **GNN** | #layer | 6 | 6 | 6 | 6 |
| | hidden dim. | 512 | 512 | 512 | 512 |
| | dropout | 0.1 | 0.1 | 0.2 | 0.2 |
| **Learning** | optimizer | Adam | Adam | SGD | SGD |
| | learning rate | 1e-4 | 1e-4 | 1e-3 | 1e-3 |
| | weight decay | 0 | 0 | 5e-4 | 5e-4 |
| | batch size | 2 | 2 | 2 | 2 |
| | # epoch | 200 | 200 | 300 | 300 |

Table 6: $F_{max}$ on EC and GO tasks under different sequence cutoffs (30% / 40% / 50% / 70% / 95%).

| Method | EC | GO-BP | GO-MF | GO-CC |
|---|---|---|---|---|
| CNN | 0.366 / 0.361 / 0.372 / 0.429 / 0.545 | 0.197 / 0.195 / 0.197 / 0.211 / 0.244 | 0.238 / 0.243 / 0.256 / 0.292 / 0.354 | 0.258 / 0.257 / 0.260 / 0.263 / 0.387 |
| ResNet | 0.409 / 0.412 / 0.450 / 0.526 / 0.605 | 0.230 / 0.230 / 0.234 / 0.249 / 0.280 | 0.282 / 0.288 / 0.308 / 0.347 / 0.405 | 0.277 / 0.273 / 0.280 / 0.278 / 0.304 |
| LSTM | 0.247 / 0.249 / 0.270 / 0.333 / 0.425 | 0.194 / 0.192 / 0.195 / 0.205 / 0.225 | 0.223 / 0.229 / 0.245 / 0.276 / 0.321 | 0.263 / 0.264 / 0.269 / 0.270 / 0.283 |
| Transformer | 0.167 / 0.173 / 0.175 / 0.197 / 0.238 | 0.267 / 0.265 / 0.262 / 0.262 / 0.264 | 0.184 / 0.187 / 0.195 / 0.204 / 0.211 | 0.378 / 0.382 / 0.388 / 0.395 / 0.405 |
| GCN | 0.245 / 0.246 / 0.246 / 0.280 / 0.320 | 0.251 / 0.250 / 0.248 / 0.248 / 0.252 | 0.180 / 0.183 / 0.187 / 0.194 / 0.195 | 0.318 / 0.318 / 0.320 / 0.323 / 0.329 |
| GearNet | 0.557 / 0.570 / 0.615 / 0.693 / 0.730 | 0.309 / 0.309 / 0.315 / 0.336 / 0.356 | 0.382 / 0.397 / 0.425 / 0.474 / 0.503 | 0.381 / 0.385 / 0.393 / 0.398 / 0.414 |
| GearNet-edge | **0.625 / 0.646 / 0.694 / 0.757 / 0.810** | **0.345 / 0.347 / 0.354 / 0.378 / 0.403** | **0.444 / 0.461 / 0.490 / 0.537 / 0.580** | **0.394 / 0.394 / 0.401 / 0.408 / 0.450** |
| DeepFRI | 0.470 / 0.505 / 0.545 / 0.600 / 0.631 | 0.361 / 0.362 / 0.371 / 0.391 / 0.399 | 0.374 / 0.383 / 0.409 / 0.446 / 0.465 | 0.440 / 0.441 / 0.444 / 0.451 / 0.460 |
| ESM-1b | 0.737 / 0.764 / 0.797 / 0.839 / 0.864 | 0.394 / 0.399 / 0.407 / 0.429 / 0.452 | **0.546 / 0.562 / 0.588 / 0.625 / 0.657** | **0.462 / 0.465 / 0.468 / 0.465 / 0.477** |
| Multiview Contrast | **0.744 / 0.769 / 0.808 / 0.848 / 0.874** | **0.436 / 0.442 / 0.449 / 0.471 / 0.490** | 0.533 / 0.548 / 0.573 / 0.612 / **0.654** | 0.459 / 0.460 / **0.467 / 0.469 / 0.488** |

2021), before GVP to additionally utilize protein sequence representations. We also adopt this hybrid model as one of our baselines, and its implementation follows the official source code.

**Our methods.** For pretraining, we use Adam optimizer with learning rate 0.001 and train a model for 50 epochs. Then, the pretrained model will be finetuned on downstream datasets.

For **Multiview Contrast**, we set the cropping length of subsequence operation as 50, the radius of subspace operation as 15, the mask rate of random edge masking operation as 0.15. The temperature $\tau$ in the InfoNCE loss function is set as 0.07. When pretraining GearNet-Edge and GearNet-Edge-IEConv, we use 96 and 24 as batch sizes, respectively.

For **Distance Prediction**, we set the number of sampled residue pairs as 256. The batch size will be set as 128 and 32 for GearNet-Edge and GearNet-Edge-IEConv, respectively. For **Residue Type**, **Angle** and **Dihedral Prediction**, we set the number of sampled residues, residue triplets and residue quadrants as 512. The batch size will be set as 96 and 32 for GearNet-Edge and GearNet-Edge-IEConv, respectively.

For downstream evaluation, the hidden representations in each layer of GearNet will be concatenated for the final prediction. Table 5 lists the hyperparameter configurations for different downstream tasks. For the four tasks, we use the same optimizer and number of epochs as in the original papers to make fair comparison. For EC and GO prediction, we use ReduceLROnPlateau scheduler with factor 0.6 and patience 5, while we use StepLR scheduler with step size 50 and gamma 0.5 for fold and reaction classification.

# F   ADDITIONAL EXPERIMENTAL RESULTS ON EC AND GO PREDICTION

**Results under different sequence identity cutoffs.** Besides the experiments in Section 5, where 95% is used as the sequence identity cutoff for EC and GO dataset splitting, we also test our models and several important baselines under four lower sequence identity cutoffs and show the experimental results in Table 6. The aim of this experiment is to test the robustness of different models under different hold-out test sets, with lowering cutoff indicating lower similarity between training and test sets. It can be observed that, at lower cutoffs, our model can still achieve the best performance among models without pretraining and get comparable or better results against ESM-1b after pretraining.

Table 7: AUPR on EC and GO prediction. [†] denotes results taken from Wang et al. (2022b). For pretraining, we select the model with the best performance when training from scratch, *i.e.*, GearNet-Edge. We omit the model name and use pretraining methods to name our pretrained models.

| | Method | Pretraining Dataset (Size) | EC | GO BP | GO MF | GO CC |
|---|---|---|---|---|---|---|
| w/o pretraining | CNN (Shanehsazzadeh et al., 2020) | - | 0.526 | 0.159 | 0.351 | 0.204 |
| | ResNet (Rao et al., 2019) | - | 0.590 | 0.205 | 0.434 | 0.214 |
| | LSTM (Rao et al., 2019) | - | 0.414 | 0.156 | 0.334 | 0.192 |
| | Transformer (Rao et al., 2019) | - | 0.218 | 0.156 | 0.177 | 0.210 |
| | GCN (Kipf & Welling, 2017) | - | 0.319 | 0.136 | 0.147 | 0.175 |
| | GAT (Veličković et al., 2018) | - | 0.320 | 0.171[†] | 0.329[†] | 0.249[†] |
| | GVP (Jing et al., 2021) | - | 0.482 | 0.224[†] | 0.458[†] | 0.279[†] |
| | 3DCNN_MQA (Derevyanko et al., 2018) | - | 0.029 | 0.132 | 0.075 | 0.144 |
| | GraphQA (Baldassarre et al., 2021) | - | 0.543 | 0.199 | 0.347 | 0.265 |
| | New IEConv (Hermosilla & Ropinski, 2022) | - | 0.775 | **0.273** | **0.572** | **0.316** |
| | **GearNet** | - | 0.751 | 0.211 | 0.490 | 0.276 |
| | **GearNet-IEConv** | - | 0.835 | 0.231 | 0.547 | 0.259 |
| | **GearNet-Edge** | - | 0.835 | 0.251 | **0.570** | 0.303 |
| | **GearNet-Edge-IEConv** | - | **0.843** | 0.244 | 0.561 | 0.284 |
| w/ pretraining | DeepFRI (Gligorijević et al., 2021) | Pfam (10M) | 0.547 | 0.282 | 0.462 | 0.363 |
| | ESM-1b (Rives et al., 2021) | UniRef50 (24M) | 0.889 | **0.332** | **0.639** | 0.324 |
| | ProtBERT-BFD (Elnaggar et al., 2021) | BFD (2.1B) | 0.859 | 0.188[†] | 0.464[†] | 0.234[†] |
| | LM-GVP (Wang et al., 2022b) | UniRef100 (216M) | 0.710 | 0.302[†] | 0.580[†] | **0.423**[†] |
| | Residue Type Prediction | AlphaFoldDB (805K) | 0.870 | 0.267 | 0.583 | 0.311 |
| | Distance Prediction | AlphaFoldDB (805K) | 0.863 | 0.274 | 0.586 | 0.327 |
| | Angle Prediction | AlphaFoldDB (805K) | 0.880 | 0.291 | 0.603 | 0.331 |
| | Dihedral Prediction | AlphaFoldDB (805K) | 0.881 | 0.304 | 0.603 | 0.338 |
| | **Multiview Contrast** | AlphaFoldDB (805K) | **0.892** | 0.292 | 0.596 | 0.336 |

**AUPR on EC and GO prediction.** We have reported experimental results on EC and GO prediction with $F_{max}$ as the metric in Section 5. Here we report another popular metric AUPR in Table 7. Note that we still use the best model selected by $F_{max}$ on validation sets. It can be observed that our model can still achieve the best performance on EC prediction in both from scratch and pretrained settings. However, there are still non-trivial gaps between our models with the state-of-the-art results. This probably is because of the inconsistency between the two evaluation metrics. It would be interesting to study the relationship between these two metrics and develop a model good at both in future works.

**Sampling schemes in self-prediction methods.** Different sampling schemes may lead to different results for self-prediction methods. We study the effects of sampling schemes using Dihedral Prediction as an example. Instead of sampling dihedral angles formed by three consecutive edges, we try to predict the dihedrals formed by four randomly sampled nodes. We observe that the $F_{max}$ decreases from $0.859$ to $0.821$. This suggests that it is better of learning residue representations to capture local spatial information instead of global information. The change of sampling schemes will make self-prediction tasks more difficult to solve, which even brings negative effects after pretraining.

**Pretraining on different datasets.** We use the AlphaFold protein structure database as our pretraining database, because it contains the largest number of protein structures and is planned to cover over 100 million proteins. However, the structures in this database are not experimentally determined but predicted by AlphaFold2. Therefore, it is interesting to see the performance of our methods when pretraining on different datasets.

To study the effects of the choice of pretraining dataset, we build another dataset using structures extracted from Protein Data Bank (PDB) (Berman et al., 2000). Specifically, we extract 123,505 experimentally-determined protein structures from PDB whose resolutions are between 0.0 and 2.5 angstroms, and we further extract 305,265 chains from these proteins to construct the final dataset.

Next, we pretrain our five methods on AlphaFold Database v1 (proteome-wide structure predictions), AlphaFold Database v2 (Swiss-Prot structure predictions) and Protein Data Bank and then evaluate the pretrained models on the EC prediction task. The results are reported in Table 8. As can be seen in the table, our methods can achieve comparable performance on different pretraining datasets. Consequently, our methods are robust to the choice of pretraining datasets.

Table 8: Results of GearNet-Edge pretrained on different pretraining datasets with different methods. Models are evaluated on the EC prediction task.

| Dataset | # Proteins | Multiview Contrast | | Residue Type Prediction | | Distance Prediction | | Angle Prediction | | Dihedral Prediction | |
|---|---|---|---|---|---|---|---|---|---|---|---|
| | | $AUPR_{pair}$ | $F_{max}$ | $AUPR_{pair}$ | $F_{max}$ | $AUPR_{pair}$ | $F_{max}$ | $AUPR_{pair}$ | $F_{max}$ | $AUPR_{pair}$ | $F_{max}$ |
| AlphaFold Database (v1 + v2) | 804,872 | 0.892 | 0.874 | 0.870 | 0.834 | 0.863 | 0.839 | 0.880 | 0.853 | 0.881 | 0.859 |
| AlphaFold Database (v1) | 365,198 | 0.890 | 0.874 | 0.869 | 0.842 | 0.871 | 0.843 | 0.879 | 0.854 | 0.877 | 0.852 |
| AlphaFold Database (v2) | 439,674 | 0.890 | 0.874 | 0.868 | 0.838 | 0.868 | 0.846 | 0.881 | 0.853 | 0.883 | 0.861 |
| Protein Data Bank | 305,265 | 0.881 | 0.859 | 0.870 | 0.841 | 0.865 | 0.847 | 0.880 | 0.857 | 0.886 | 0.858 |

## G   STRUCTURE PRETRAINING ON EGNN

To verify the effectiveness of our proposed methods, we choose another common backbone model, equivariant graph neural network (EGNN) (Satorras et al., 2021), for pretraining. We follow the experimental setup in Appendix E.3 for pretraining and finetuning the model. The results on the EC dataset are reported in Table 9. It can be seen that the performance of the EGNN is improved by a large margin with all the five pretraining methods. Among them, Distance Prediction and Multiview Contrast are the top two methods. The ranks of these five methods are quite different from the ranks in the main paper. This is probably because the capacity of EGNN limits its performance on some pretraining tasks and thus reduces their benefits.

| Method | $F_{max}$ |
|---|---|
| **EGNN** | 0.640 |
| Residue Type Prediction | 0.729 |
| Distance Prediction | **0.761** |
| Angle Prediction | 0.718 |
| Dihedral Prediction | 0.662 |
| **Multiview Contrast** | 0.752 |

Table 9: Pretraining results on EC with EGNN as backbone models.

## H   COMBINE SEQUENCE- AND STRUCTURE-BASED ENCODERS

In the main paper, we compare the pretrained sequence- and structure-based encoder and show that geometric structure pretraining can achieve competitive results with much less data. To further show the benefit of incorporating structural information, here we choose ESM-1b as a baseline and build a structure-based encoder based on its output. Specifically, we replace the raw node features in GearNet with pretrained sequence representations. We train the model on EC with the same configurations for GearNet and finetune the ESM-1b model with learn-

| Method | EC | GO | | |
|---|---|---|---|---|
| | | BP | MF | CC |
| GearNet | 0.730 | 0.356 | 0.503 | 0.414 |
| GearNet-Edge | 0.810 | 0.403 | 0.580 | 0.450 |
| -w/ Multiview Contrast | 0.874 | 0.490 | 0.654 | 0.488 |
| ESM-1b | 0.864 | 0.452 | 0.657 | 0.477 |
| **ESM-1b+GearNet** | **0.883** | **0.491** | **0.677** | **0.501** |

Table 10: Results ($F_{max}$) for combining seuqence- and structure-based encoders.

ing rate 1e-5. The results are shown in Table 10. It can be observed that the ESM-1b+GearNet model can achieve SOTA performance even without structure-based pretraining, which suggests the importance of utilizing protein structures. Also, it is promising to explore pretraining methods on the combined encoder. We leave this direction for future work.

## I   COMBINE NEURAL AND RETRIEVAL-BASED METHODS

Searching a database to retrieve similar proteins is a popular method used in the biological community when predicting properties of a target protein, *e.g.*, searching multiple sequence alignments and templates for structure prediction (Jumper et al., 2021). There have been a large amount of tools proposed for aligning protein sequences (Altschul et al., 1997; Steinegger & Söding, 2017; Steinegger et al., 2019) and structures (Yang & Tung, 2006; Zhang & Skolnick, 2005; van Kempen et al., 2022; Holm, 2019). In this section, we first compare our neural representation-based method with retrieval-based methods on function and fold classification tasks and then showcase the potential of our proposed methods on structure-based search tasks.

**Comparison with retrieval-based methods.**   We select a structure alignment tool, Foldseek (van Kempen et al., 2022), as our retrieval-based baseline. The method trains a VQ-VAE on SCOPe40 to discretize structural units into an alphabet of twenty 3Di states and then transforms the problem to 3Di sequence alignment, which is done by MMseqs2 (Steinegger & Söding, 2017). When using

| Method | EC | | GO-BP | | GO-MF | | GO-CC | | Fold Classification | | | |
|---|---|---|---|---|---|---|---|---|---|---|---|---|
| | $AUPR_{pair}$ | $F_{max}$ | $AUPR_{pair}$ | $F_{max}$ | $AUPR_{pair}$ | $F_{max}$ | $AUPR_{pair}$ | $F_{max}$ | Fold | Super. | Fam. | Avg. |
| Foldseek (van Kempen et al., 2022) | 0.778 | 0.888 | 0.168 | 0.440 | 0.462 | 0.649 | 0.159 | 0.321 | 2.78 | 7.57 | 65.4 | 25.2 |
| **GearNet-Edge(-IEConv)** | 0.835 | 0.810 | 0.251 | 0.403 | 0.570 | 0.580 | 0.303 | 0.450 | 48.3 | 70.3 | 99.5 | 72.7 |
| **w/ Multiview Contrast** | 0.892 | 0.874 | 0.292 | 0.490 | 0.596 | 0.654 | **0.336** | **0.488** | **54.1** | **80.5** | 99.9 | **78.1** |
| **w/ Multiview Contrast + Foldseek** | **0.908** | **0.903** | **0.314** | **0.500** | **0.615** | **0.673** | 0.319 | 0.467 | - | - | - | - |

Table 11: Comparison between neural and retrieval-based methods on EC, GO and fold classification tasks. As in Table 2, we use GearNet-Edge on EC and GO prediction and GearNet-Edge-IEConv on fold classification as backbone models. The results w/o and w/ pretraining and those ensembled with Foldseek are reported on EC and GO. We omit the ensemble results on fold classification due to the poor performance of Foldseek.

Foldseek, we follow the parameters provided in their github repo[2]. For each protein in the test set of our benchmark tasks, we use Foldseek to retrieve the most similar protein in the training set, the label of which will be used for prediction.

We report the results of retrieval-based and our proposed neural methods in Table 11. First, we find that Foldseek achieves very good performance on EC and GO prediction. In terms of $F_{max}$, it is better than GearNet-Edge on all tasks and competitive with the pretrained GearNet-Edge on EC and GO-MF. The accuracy of Foldseek makes it a strong baseline for proteins with similar structures in the training set. However, when the dataset is split by structural similarities, *e.g.*, fold classification, the structural alignment tool fails to retrieve similar proteins and get accurate prediction as shown in the table. This can be attributed to the inherent limitation of retrieval-based methods, *i.e.*, the lack of generalization ability to novel data points.

Furthermore, to utilize the advantages from both worlds, we combine neural and retrieval-based methods via ensemble. As shown in the last row of Table 11, both metrics are significantly improved on EC, GO-BP, GO-MF compared with the separate neural and retrieval-based methods. Consequently, it would be interesting to explore the combination of these two kinds of methods in the future, as have done in many machine learning tasks (Mitra & Craswell, 2017; Sun et al., 2019; Notin et al., 2022).

**Results on structure-based search tasks.** To show the potential of structure-based modeling for biological applications, we test our model on the SCOPe40 benchmark proposed in van Kempen et al. (2022). The authors cluster the SCOPe 2.0198 at 40% sequence identity and obtain 11,211 non-redundant protein sequences. They perform an all-versus-all search on the dataset and test the ability of structure alignment tools for finding proteins of the same SCOPe family, superfamily, and fold. For each query, they measure the fraction of TPs (true positive matches) out of all correct matches until the first FP (false positive) that matches to a different fold. The sensitivity is calculated by the area under the curve of the cumulative ROC curve up to the first FP.

Table 12: Sensitivity of searching proteins of the same family, superfamily and fold on SCOPe40. Results are evaluated with the scripts and predictions provided in (van Kempen et al., 2022).

| Method | Fold | Super. | Fam. | Avg. |
|---|---|---|---|---|
| MMseqs2 (Steinegger & Söding, 2017) | 0.001 | 0.082 | 0.542 | 0.208 |
| 3D-BLAST (Yang & Tung, 2006) | 0.009 | 0.126 | 0.572 | 0.235 |
| CLE-SW (Zhao et al., 2013) | 0.021 | 0.293 | 0.763 | 0.359 |
| CE (Shindyalov & Bourne, 1998) | 0.131 | 0.529 | 0.885 | 0.515 |
| TMalign-fast (Zhang & Skolnick, 2005) | 0.188 | 0.618 | 0.906 | 0.571 |
| TMalign (Zhang & Skolnick, 2005) | 0.188 | 0.610 | 0.901 | 0.566 |
| Foldseek (van Kempen et al., 2022) | 0.155 | 0.593 | 0.914 | 0.554 |
| DALI (Holm, 2019) | 0.310 | 0.751 | 0.942 | 0.667 |
| **GearNet-Edge-IEConv** | 0.474 | 0.722 | 0.936 | 0.710 |
| **GearNet-Edge-IEConv + DALI** | **0.481** | **0.779** | **0.963** | **0.741** |

---

[2]https://github.com/steineggerlab/foldseek-analysis/blob/main/scopbenchmark/scripts/runFoldseek.sh

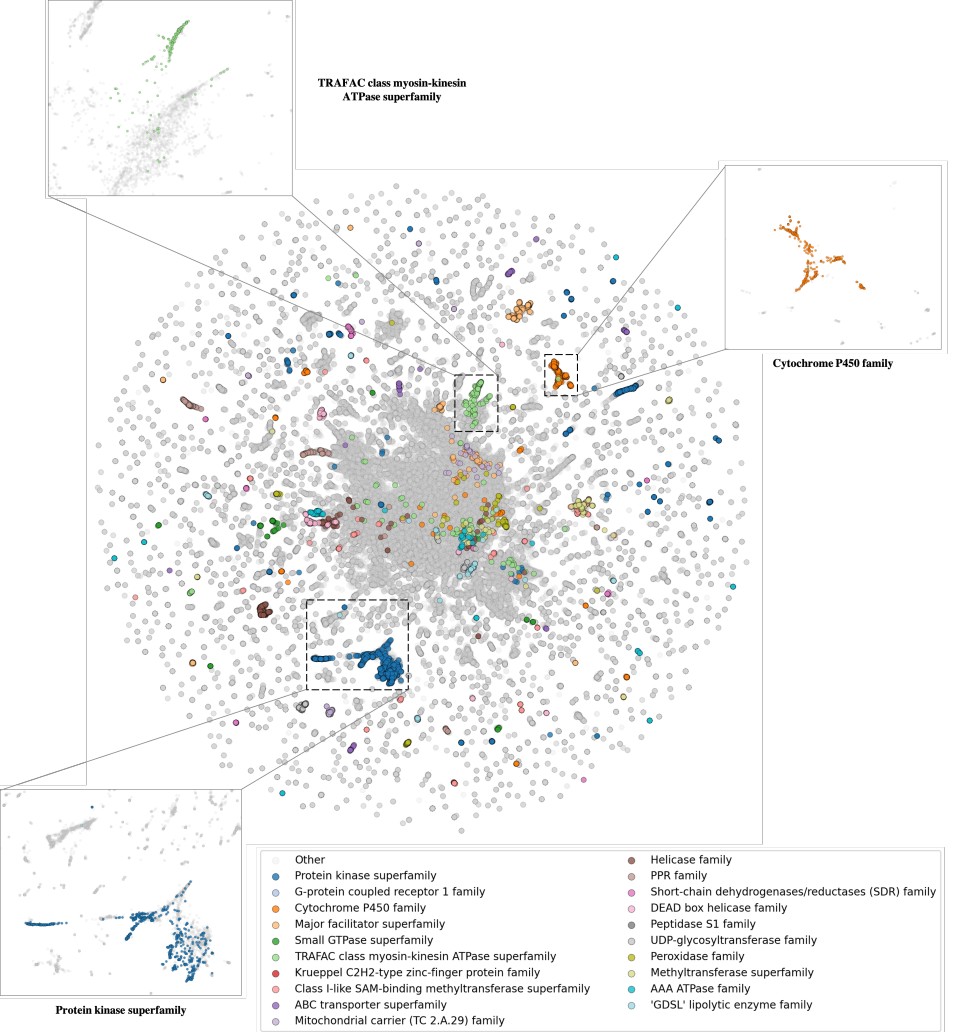

Figure 6: Latent space visualization of GearNet-Edge (Multiview Contrast) on AlphaFold Database v1.

We report the results of seven structure alignment tools and a sequence search tool evaluated in van Kempen et al. (2022). For comparison, we use the GearNet-Edge-IEConv model trained on fold classification to extract representations for proteins and use the cosine similarity between representations to retrieve similar proteins. The sensitivity of these methods is reported in Table 12. It can be observed that our method achieves the best performance on average among all baselines. Compared with DALI, though finding fewer proteins of the same family and superfamily, our method can achieve higher sensitivity at the fold level. This can be explained by the better generalization ability to novel structures of neural methods. With the ensemble of neural and retrieval-based methods, we can achieve the SOTA performance at all levels. This again demonstrates the effectiveness of our proposed method and the potential of combining neural and retrieval-based methods.

## J    LATENT SPACE VISUALIZATION

For qualitatively evaluating the quality of the protein embeddings learned by our pretraining method, we visualize the latent space of the GearNet-Edge model pretrained by Multiview Contrast. Specifically, we utilize the pretrained model to extract the embeddings of all the proteins in AlphaFold Database v1, and these embeddings are mapped to the two-dimensional space by UMAP (McInnes et al., 2018) for visualization. Following Akdel et al. (2021), we highlight the 20 most common super-

**1NYR-A (ATP Binding)**    **1B85-A (Heme Binding)**

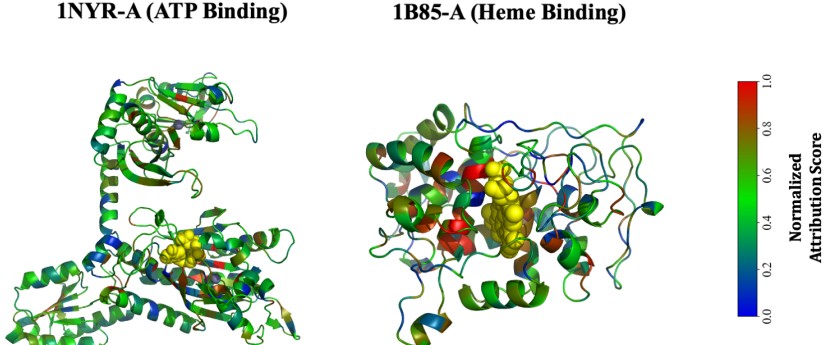

Figure 7: Identification of active sites on proteins responsible for binding based on attribution scores. Two proteins binding to specific targets are selected for illustration (1NYR-A for ATP binding and 1B85-A for Heme binding). For these two complexes, ligands are shown in yellow spheres while the residues of the receptors are colored based on attribution scores. Residues with higher attribution scores are colored in red while those with lower scores are colored in blue.

families within the database by different colors. The visualization results are shown in Fig. 6. It can be observed that our pretrained model tends to group the proteins from the same superfamily together and divide the ones from different superfamilies apart. In particular, it succeeds in clearly separating three superfamilies, *i.e.*, Protein kinase superfamily, Cytochrome P450 family and TRAFAC class myosin-kinesin ATPase superfamily. Such a decent capability of discriminating protein superfamilies, to some degree, interprets our model's superior performance on Fold Classification.

## K    RESIDUE-LEVEL EXPLANATION

Protein functions are often reflected by specific regions on the 3D protein structures. For example, the binding ability of a protein to a ligand is highly related to the binding interface between them. Hence, to better interpret our prediction, we apply Integrated Gradients (IG) (Sundararajan et al., 2017), a model-agnostic attribution method, on our model to obtain residue-level interpretation. Specifically, we first select two molecular functions, ATP binding (GO:0005524) and Heme binding (GO:0020037), from GO terms that are related to ligand binding. For each functional term, we pick one protein and feed it into the best model trained on the GO-MF dataset. Then, we use IG to generate the feature attribution scores for each protein. The method will integrate the gradient along a straight-line path between a baseline input and the original input. Here the original input and baseline input are the node feature $f$ and a zero vector, respectively. The final attribution score for each protein will be obtained by summing over the feature dimension. The normalized score distribution over all residues are visualized in Figure 7. As can be seen, our model is able to identify the active sites around the ligand, which are likely to be responsible for binding. Note that these attributions are directly generated from our model without any supervision, which suggests the decent interpretability of our model.

