# OpenReview forum: "Protein Representation Learning by Geometric Structure Pretraining"
_ICLR.cc/2023/Conference — ICLR 2023 poster_

### Official Review · Reviewer_roE6 · 2022-10-22

**Confidence:** 4
**Correctness:** 4
**Technical Novelty And Significance:** 3
**Empirical Novelty And Significance:** 3
**Recommendation:** 8

**Clarity, Quality, Novelty And Reproducibility:**

Clarity: The paper is clearly written and easy to follow.

Quality: The paper is technically sound. Evaluations are correctly implemented with proper train/test splits. The equivariant model is equivariant.

Novelty: The paper is not that novel. On the modeling side, equivariance is achieved by ensuring the model only operates on equivariant features, which is a common approach. No major new capabilities (modeling side chains, handling uncertain inputs) is discussed. On the pretraining side, Hsu et al. 2022 did perform large scale structure-based pretraining in the context of an inverse-folding model and showed significant benefit.

Reproducibility: The results seem reproducible. Previously published datasets and splits are used. The authors promise to release code + model weights upon publication.

**Strength And Weaknesses:**

Strengths:
* The paper is technically sound, and the method presented is relatively simple and scalable, so could be extended to training on larger datasets relatively easily.
* The paper is well written and easy to follow + understand
* Baselines chosen are reasonable and convincing that this method performs well on the chosen downstream tasks.

Weaknesses:

1) The downstream evaluations chosen are just not that convincing to me. This set of tasks has become somewhat standard among a subset of the geometric modeling community, but these tasks are fairly contrived, and the results are not so impressive.
    - 3D structure-based models are *provided the structure as input*. This is a huge amount of information relative to sequence-based models (as the authors state in the paper). So I am simply confused as to why they perform only slightly better than sequence based models such as ESM-1b. I would expect that, given the structure, it would be very simple to outperform models such as ESM-1b for many function-prediction based tasks, especially since ESM-1b likely contains less information than a multiple sequence alignment.
    - Before breaking out a neural network predictor, the approach I would take to classify functions based on structure would be to use a structure search tool, such as foldseek. A simple baseline therefore would be a 1-NN algorithm: take the test set structure, use foldseek to search the training set, and return the label of the structure with best similarity. Is the approach proposed here better than that baseline?
    - The one task where structure-based methods show a clear advantage is the fold classification task. However, the construction of this task deliberately designed to advantage structure-based models over sequence-based models. The dataset splits were constructed such that the goal was to match inputs with very different sequences, but similar structures (i.e. remote homology detection). When the structure is provided, however, it is not clear why this task is interesting. I would expect the baseline I suggested above to perform very well.

2) There are two application areas to my knowledge where structure-based modeling has shown a great deal of promise. The first is inverse folding (e.g. Dauparas et al. 2022, https://www.science.org/doi/10.1126/science.add2187), which has enabled breakthroughs in de novo protein design. The second is structure-based search (van Kempen et al. 2022, https://www.biorxiv.org/content/10.1101/2022.02.07.479398v4) which uses a small neural network to speed up structural searches, an important task given the development of large scale protein structure databases.

The second task (structure-based search) seems like a very promising application for the model, and would be much more convincing than those shown in the paper. Given the contrastive nature of the training method, structural similarity search seems like a natural application (although care would have to be taken to ensure generalization to novel structures outside the method's training set).

3) The inability to model side chains seems like a potential drawback for structure-based methods. Especially in regards to function / binding affinity prediction, this may hinder the ability to model the output better than sequence-based models can.

Minor Comment: How much does the use predicted structures affect the model's capabilities? What is the relationship e.g. between AlphaFold pLDDT and model accuracy if predicted structures are used?

**Summary Of The Paper:**

The paper introduces a simple SE(3) invariant model and pretrains it on large dataset of alphafold predicted structures. This is then finetuned on a handful of downstream tasks, where it achieves better results than some prior methods.

**Summary Of The Review:**

The paper is technically sound and is likely an improvement on the prior methods they benchmark against (e.g. Hermosilla et al). However, the paper does not add major novel insights, and does not have strong results on actually interesting downstream tasks.

I recommend the authors look into areas where structure-based modeling is actually used in practice and seeing how their model may be applied.

*********** After Author Response ***********

The authors have performed significant additional work, including adding the baseline I requested (kNN-1 search via Foldseek) and a benchmark on structure search as a task. The results on this are very promising since fast structure-based search is actually a current pressing issue in the field. Combining accurate cosine-similarity based search with tools like FAISS is potentially a highly valuable tool for the community, and something I would encourage the authors to pursue.

Overall, I now believe this paper is a good candidate for acceptance at ICLR and have revised my score.

---

> ### Author Response · Authors · 2022-11-13
> **Author Feedback to Reviewer roE6 (1/3)**
>
>
> Thanks for your constructive suggestions! We have enhanced our experiments with your suggested baselines and tasks in Appendix H & I of the latest version. Here is our response to your concerns.
>
> >**Q1: Why do structure-based models perform only slightly better than sequence-based models such as ESM-1b?**
>
> First, ESM-1b is a very strong baseline on function prediction tasks as proved in [a], even better than the MSA-based and structure-based models. This is because **huge models with large-scale pretraining can achieve outstanding performance, even with less information in the input.** This has already been demonstrated by many famous works in the machine learning community [b,c,d,e]. For comparison, there are 650M learnable parameters in ESM-1b v.s. 42M in GearNet-Edge. Also, the pretraining dataset used for ESM-1b consists of 24M proteins v.s. 805K for GearNet-Edge. We believe that our structure-based models can achieve much better performance when scaling to the full AlphaFold Database.
>
> Then, to answer your question, we add experiments in Appendix H to show the effect of incorporating structural information. We build structure-based models upon pretrained ESM-1b representations. As shown in the paper, **the ESM-1b representations can be improved significantly after feeding into our GearNet encoder.** The model can achieve SOTA performance even without structure-based pretraining.
>
> Therefore, it is a very promising direction to study how to combine sequence- and structure-based models and how to do pretraining on these models. However, the main focus of this paper is the structure-based encoder and pretraining methods. We’d like to leave this direction for future work.
>
> [a] Exploring evolution-based & -free protein language models as protein function predictors. Hu et al., NeurIPS, 2022.
>
> [b] Language Models are Few-Shot Learners. Brown et al., 2020.
>
> [c] Learning Transferable Visual Models From Natural Language Supervision. Radford et al., 2021.
>
> [d] Image as a Foreign Language: BEiT Pretraining for All Vision and Vision-Language Tasks. Wang et al., 2022.
>
> [e] An Image is Worth 16x16 Words: Transformers for Image Recognition at Scale. Dosovitskiy et al., ICLR, 2021.
>
> >**Q2: A simple baseline would be a 1-NN algorithm: take the test set structure, use foldseek to search the training set, and return the label of the structure with best similarity. Is the approach proposed here better than that baseline?**
>
> Thanks for proposing this simple but very useful baseline! We’ve added detailed experimental comparison with these retrieval-based methods in Appendix I of the revised version. In summary, *the baseline achieves very good performance on function prediction tasks but still not so good as our pretrained model*. When the dataset is split by structural similarities, e.g., fold classification, **the inherent limitation of retrieval-based methods makes it difficult to generalize to novel structures**.
>
> Nevertheless, it is still promising to **combine neural and retrieval-based methods** to utilize the advantages from both worlds i.e., **the accuracy of retrieval-based methods and the generalization ability of neural methods**. As shown in the experiments in Appendix I, a simple ensemble of the two kinds can yield a SOTA model.
>
> **Though the retrieval-based methods are simple and commonly used in the protein community, this should not diminish the value of our work on the development of neural methods given their good generalization ability.** Besides, there are a series of works on combining neural and retrieval-based methods with strong performance in information retrieval [f], question answering [g] and protein engineering tasks [h]. Therefore, we believe that this will be a very interesting future direction with great potential.
>
> [f] Neural Models for Information Retrieval. B Mitra and N Craswell, 2017.
>
> [g] Pullnet: Open domain question answering with iterative retrieval on knowledge bases and text. Sun et al., EMNLP, 2019.
>
> [h] Tranception: protein fitness prediction with autoregressive transformers and inference-time retrieval. Notin et al, ICML, 2022.

---

> ### Author Response · Authors · 2022-11-13
> **Author Feedback to Reviewer roE6 (2/3)**
>
> >**Q3: The one task where structure-based methods show a clear advantage is the fold classification task. However, the construction of this task deliberately designed to advantage structure-based models over sequence-based models. The dataset splits were constructed such that the goal was to match inputs with very different sequences, but similar structures (i.e. remote homology detection). When the structure is provided, however, it is not clear why this task is interesting. I would expect the baseline I suggested above to perform very well.**
>
> First, structure-based methods also achieve strong performance on function prediction tasks given the observation that **they can achieve better performance than sequence-based models with much less pretraining data**. Also, the new experiments in Appendix H prove the huge potential of incorporating structural information over pretrained sequence representations.
>
> Second, we admit that structure-based models have advantages over sequence-based ones in fold classification tasks. Nevertheless, **the aim of including the fold classification task here is to test which machine learning method can extract meaningful representations capturing structural information**. From this perspective, the superior performance of our pretrained models over previous methods on this task is of great value for the machine learning community as a state-of-the-art protein structure representation method. It should also be noted that the representations extracted by machine learning encoders can be used in the searching method for computing the similarity between protein structures. Moreover, **the similarity scores computed by our model can be combined with the ones computed by traditional methods for better searching performance**. These have been demonstrated by experiments on structure-based search tasks in Appendix I.
>
> Third, we have added the comparison with the Foldseek baseline in Appendix I. It can be observed that its performance is much worse than our model. This is probably because the VQ-VAE in Foldseek are trained on SCOPe to match proteins within the same family and thus not good at searching proteins in the same superfamily and fold. This phenomenon is also observed in the structure-based search benchmark.
>
> >**Q4: There are two application areas to my knowledge where structure-based modeling has shown a great deal of promise. The first is inverse folding, which has enabled breakthroughs in de novo protein design. The second is structure-based search which uses a small neural network to speed up structural searches, an important task given the development of large scale protein structure databases.**
>
> Thanks for bringing these two tasks to our attention! We have added the structure-based search benchmark in Appendix I and proved the usefulness of our methods. This also enlightens us for exploring the combination of neural and retrieval-based methods in the future.
>
> As for inverse folding, we agree that this is a very important and fundamental topic in protein science. However, the problem definition is quite different from what we study in the paper. In this work, we focus on **encoding protein sequences and structures into meaningful representations reflecting protein functional and structural properties**, whereas **the inverse folding problem aims to predict the protein sequence given the protein structure**. Although *Hsu et al. 2022 did perform large scale structure-based pretraining in the context of an inverse-folding model*, the problems and contexts are very different from ours.
>
> >**Q5: The inability to model side chains seems like a potential drawback for structure-based methods. Especially in regards to function / binding affinity prediction, this may hinder the ability to model the output better than sequence-based models can.**
>
> Thanks for the question! We admit that side chain information is very important for complex tasks, e.g., protein-protein interaction, protein-ligand binding. In this work, we find that simply keeping CAs can achieve good performance on the considered tasks, since function and fold prediction only need coarser-level representations. Nevertheless, **our model can be easily generalized to protein side-chain modeling by extending the residue graphs to atom level**. Specifically, we can choose which atoms to keep for each protein (CA, backbones or side-chain atoms) and construct different types of edges between atoms (sequential, radius, knn or bond edges). In this way, we can explicitly model the interactions between atoms and thus enable the prediction for more complicated tasks, e.g., binding affinity prediction. It would be also interesting to explore different ways of constructing atom graphs, so we leave this for future work.

---

> ### Author Response · Authors · 2022-11-13
> **Author Feedback to Reviewer roE6 (3/3)**
>
> >**Q6: How much does the use of predicted structures affect the model's capabilities? What is the relationship e.g. between AlphaFold pLDDT and model accuracy if predicted structures are used?**
>
> This is a good point. In Appendix F, we have performed experiments to study the effect of different pretraining databases. We find that **switching from the predicted structures (AlphaFold DB) to accurate structures (PDB) does not have large influence on pretraining methods.** The relationship between AlphaFold pLDDT and model accuracy, to the best of our knowledge, has largely been unexplored in the literature. This is an important topic to understand the behavior of pretraining. However, it is far from the topic of this paper and will take too many resources to study. We’d like to leave this as future work.
>
> >**Q7: The paper is not that novel. On the modeling side, equivariance is achieved by ensuring the model only operates on equivariant features, which is a common approach.**
>
> We respectfully disagree with the reviewer on this point. Although there are many existing invariant encoders for molecules, their adaptation on proteins is non-trivial due to the size and different nature of proteins. In this work, **we find that simply constructing protein graphs with different edge types and modeling with different kernel matrices works very well.** Based on this novel observation, we propose a straightforward and effective encoder. This explains the novelty underlying our model and we believe the simpleness of our method will make it suitable for downstream applications.
>
> >**Q8: The downstream evaluations chosen are just not that convincing to me. This set of tasks has become somewhat standard among a subset of the geometric modeling community, but these tasks are fairly contrived, and the results are not so impressive. I recommend the authors look into areas where structure-based modeling is actually used in practice and seeing how their model may be applied.**
>
> Thanks for your suggestion! We believe that the added experiments in the revised version can address your concerns about evaluations and performance to some extent.
>
> We want to argue that different audiences may appreciate different types of downstream tasks. **The tasks we choose are standard, meaningful and clearly defined in the sense of providing clean datasets, splits and competing baseline results, which are good for evaluating machine learning models.** There are still many unstudied but interesting tasks on proteins, e.g., protein-protein interaction, protein-ligand binding, protein engineering tasks and so on. But we cannot expect one conference paper to solve all these important tasks.
>
> To emphasize, we think that the large amounts of experiments done in this paper (12 tables, 7 figures, 10+ pages of supplementary materials) are sufficient to support our claims about structure-based methods and set up a solid starting point for geometric structure pretraining. Further, the additional experiments in the appendix envision some promising future directions for solving the interesting tasks mentioned above.

---

> > ### Comment · Reviewer_roE6 · 2022-11-21
> > **Revised score to accept**
> >
> > Thank you for the significant work you have performed to address my concerns. I have revised my score to an accept and edited the review summary with additional comments.

---

### Official Review · Reviewer_86an · 2022-10-23

**Confidence:** 3
**Correctness:** 3
**Technical Novelty And Significance:** 3
**Empirical Novelty And Significance:** 3
**Recommendation:** 8

**Clarity, Quality, Novelty And Reproducibility:**

I think the quality of the work is quite high. There are always holes (e.g. no side chain information being used), especially as there are unlimited valid ways to evaluate the quality of a "representation", but given the amount of different contributions and results, it would be unreasonable to request more for an ICLR paper.
Maybe this work would be a better fit for a journal offering a longer format?
In my opinion none of the contributions presented is ground-breaking in itself but that does not need to be a criticism. The value of this work lies in putting together the right pieces together (hence the need for justifying the choices) and performing a relatively deep experimental validation.
The clarity is, the part that I think could be more easily improved. Please refer to the points I raised in the previous section of the review for more detailed suggestions.
The reproducibility of the work is currently hindered by the code not being accessible. This being said, the authors have committed to making it available upon acceptance and provide enough details about the architecture and (numerous!) hyperparameters for the motivated reader to try and reproduce the result.

**Strength And Weaknesses:**

Strength:
- Protein modeling / representation is indeed a very important topic that has a high impact both inside and outside biomedical applications. By tackling this problem in a relatively generic way (the representations are not tied to a specific downstream task), this paper definitely has a very high potential impact.
- The suggested approach seems to be well "rooted" in the current trends in the field. Recent breakthroughs offered by AlphaFold2 to (mostly) accurately predict protein structure definitely offer new opportunities for designing methods that leverage 3D structure information.
- To my knowledge, the related work section is well documented and it is relatively clear where the authors' contributions are.
- The experimental section of the paper is quite elaborate with GearNet* being evaluated on various tasks, datasets and against a relatively large number of competing methods and baselines.
- The amount of work seems substantial, to the point where the authors provide 10 pages of supplementary material with a lot of interesting comments and results.

Weaknesses:
- While the idea of having a better general protein representation (as opposed to one tailored for a specific downstream task) is of course seducing, it makes for a much stronger claim that is hard to completely substantiate. The results seem good on various downstream tasks but it is of course not clear how that would carry over other relevant tasks (for instance anything pertaining to predicting protein-protein interactions).
- While the paper reads well overall, I find that some parts lack clarity. In general, many choices of modeling could be better justified and/or offer more insights.
- Notably, from reading the "Protein graph construction" subsection, I was confused about which type of features where used for the nodes. Only the spatial coordinates are mentioned, while not actually used and I had to go deep into the supplements to really understand that the type of residue, e.g. the actual sequence information was directly used.
- I find Section 3.2 to also lack clarity. Can the authors offer a bit more insight on what kind of information the angles, that determine the edge types, actually give? Can you describe more clearly how it's different (and why a better choice) from the representation used in AlphaFold2?
- Why are those angles (in section 3.2, but also in 4.2) discretized? Would it not be a more straightforward option to keep the angles as float values (and perform regression in 4.2)?
- Despite some efforts, I do not really understand the assumption underlying the contrastive learning approach. Especially for proteins with longer and less redundant sequences, why should different views, describing potentially radically different regions of the protein, be more related to each other than views taken from different proteins?
- The construction of the views, mixing subspace and subsequence sampling, along with the addition of masking or not, seems a bit convoluted to me. From reading the manuscript, I don't get a strong intuition why this mixture is meaningful and the results reported in Table 3 seem to indicate that adding the subsequence sampling and identity transformation does not yield better performance.
- I am not sure I really like the argument that the proposed method pre-trains with much less data than sequence-based approaches. While the number of proteins used for pre-training is indeed much lower, much more data per protein is being used when incorporating 3D structure.
- Related to the previous point, it would be nice to see some runtime and memory usage analysis of the different methods used.
- Although it is qualitative, and not entirely compelling because the analysis has been performed only for the author's model, and it could be that they are hand picked, I find the results presented in Appendix H to be very interesting. It would be nice to at least put a reference to it in the main text.

More open question:
The current model includes mostly structural information about the backbone of the protein, which seems to be sufficient for the chosen downstream tasks. For tasks like complex interaction prediction, that probably will be a bottleneck.
Do the authors have suggestion as to how to extend the current model to include side chain information?

Minor:
- Enzyme Commission acronym EC is used in section 4.1 but only defined later in 5.1.
- There's a typo just before 4.2 "subsequecne".

**Summary Of The Paper:**

While a large part of the literature focuses on learning protein representation from their amino acid sequences (allowing pre-training from huge existing database of known protein sequences), the authors suggest a novel approach to learn representation from their 3D structure.
The first contribution lies in the definition of a novel encoder called GearNet, based on a relational Graph Neural Network.
The protein at hand is initially represented as a graph, where each node (i.e. the alpha carbon of a residue) is characterized by its 3D position in its observed or predicted structure.
The edges are added based on closeness in the sequence or in an Euclidean sense (nodes within a certain radius and K-nearest neighbor nodes), if they are not close in a sequence sense.
To obtain the protein representation, the aforementioned graph is passed through a relational GNN.
The authors suggest a second contribution to explicitly model interactions between edges of the graph: they design an edge message passing layer, coined GearNet-Edge.
It is based on a relational graph where the nodes are the edges of the aforementioned graph and the edges in this new graph represent the (discretized) angles between the adjacent edges in the previous graph.
A straightforward message passing layer is defined over that new graph and the aggregation function in GearNet-Edge combines the aggregation functions defined over both graphs.
As another contribution, the authors suggest to pre-train the resulting encoders on large collections of unlabeled protein structures.
To do so, they suggest a procedure inspired from SimCLR where different, random, views of a protein graph are created by cropping their full graph and randomly masking some of the edges.
Pairs of views are generated and labeled as positive if they originate form the same initial protein.
They also suggest more straightforward baselines to pre-train their encoders based on self-prediction where the encoders are used to perform masked predictions (residue type, distances etc.) on single, pair, triplets or quadruplets of residues.
Finally, the authors validate the approach by comparing their (pre-trained or not) method with different baselines and sequence and structure-based approaches from the literature, on 4 different downstream prediction tasks: enzyme commission number, gene ontology term, fold classification, reaction classification.
They also perform an ablation study to validate the different contributions more individually.

**Summary Of The Review:**

Overall, the authors suggest a novel method to learn efficient representations of proteins, leveraging their 3D structure.
Following the aftermath of the recent release of AlphaFold2 and their prediction of the structure of a huge amount of new data, this contribution seems well-timed and seem to result in good performance compared to other very recent methods.
I hope that this review process will offer the authors the opportunity to especially improve the clarity of the current manuscript.
That being said, I believe that time is of the essence as I suspect that a handful of related methods are currently being developed and published.
Given the general quality, I recommend that this paper be accepted.

---

> ### Author Response · Authors · 2022-11-13
> **Author Feedback to Reviewer swLM (1/2)**
>
> Thanks for your suggestions! We have improved the clarity of our paper following your suggestions. Here is our response to your concerns.
>
> >**Q1: While the idea of having a better general protein representation is of course seducing, it makes for a much stronger claim that is hard to completely substantiate. The results seem good on various downstream tasks but it is of course not clear how that would carry over other relevant tasks.**
>
> Thanks for pointing out this! We have toned down our claim in the introduction of the revised version.
>
> >**Q2: From reading the "Protein graph construction" subsection, I was confused about which type of features where used for the nodes.**
>
> Thanks! We have added a sentence in Sec. 3.1 to describe the node features in the main paper.
>
> >**Q3: Can the authors offer a bit more insight on what kind of information the angles, that determine the edge types, actually give? Can you describe more clearly how it's different (and why a better choice) from the representation used in AlphaFold2? Why are those angles (in section 3.2, but also in 4.2) discretized? Would it not be a more straightforward option to keep the angles as float values (and perform regression in 4.2)?**
>
> Thanks for the questions! We have explained the intuitions in the revised version.
>
> For edge message passing, the **angular information reflects the relative position between two edges that determines the strength of their interaction**. **The angles are discretized to save memory costs** for computing a large number of kernel matrices. The reason is the same as that for replacing edge-wise kernel matrices in IEConv with relational kernel matrices in GearNet. Compared with the triangle attention in AlphaFold2, our method considers angular information to model different types of interactions between edges, which are more efficient for sparse edge message passing.
>
> Table A: F_max on EC for pretraining methods with regression or classification loss.
>
> |#Method|Regression|Classification|
> |:----:|:----:|:----:|
> |Distance Prediction|0.830|0.829|
> |Angle Prediction|0.830|0.853|
> |Dihedral Prediction|0.839|0.859|
>
> **For angle prediction, since angular values are more sensitive to errors in protein structures than distances, we use discretized values for prediction.** Also, we report the results for self-prediction baselines pretrained with regression and classification losses in the table above. It can be seen that using classification losses is the better choice for angle and dihedral prediction.
>
> >**Q4: Despite some efforts, I do not really understand the assumption underlying the contrastive learning approach. Especially for proteins with longer and less redundant sequences, why should different views, describing potentially radically different regions of the protein, be more related to each other than views taken from different proteins?**
>
> The essential intuition behind contrastive learning is that representations of substructures from the same protein are likely to be more similar than those from different proteins. Similar ideas have been used for learning image and graph representations. For example, there are also cropping and resizing operations in SimCLR [a] and MoCo v2 [b]. **During pretraining, we employ the proteins from AlphaFold Database containing only single-chain proteins.** We admit that **there may be some long proteins** that contain structurally different motifs. However, **these cases are very rare** compared with the cases where augmentation functions can capture related substructures. Just like a few images that contains cats and dogs (different animals) would not affect the whole training of the pretraining algorithm [a, b].
>
> [a] A simple framework for contrastive learning of visual representations. Chen et al., ICML, 2020.
>
> [b] Improved Baselines with Momentum Contrastive Learning. Chen et al., 2020.
>
> >**Q5: The construction of the views, mixing subspace and subsequence sampling, along with the addition of masking or not, seems a bit convoluted to me. From reading the manuscript, I don't get a strong intuition why this mixture is meaningful and the results reported in Table 3 seem to indicate that adding the subsequence sampling and identity transformation does not yield better performance.**
>
> This is a good point. In the current version, we add more results on GO for this ablation study. According to Table 3, **the benefits of randomly sampling view functions in our method can be clearly observed on all three tasks the GO dataset.** This can be understood since generating diverse view functions is very important for contrastive learning [a].
>
> [a] A simple framework for contrastive learning of visual representations. Chen et al., ICML, 2020.

---

> > ### Comment · Reviewer_86an · 2022-11-16
> > **Thanks for the careful answers**
> >
> > Congratulations on integrating new meaningful results, revising the manuscript (and still fitting in the 9 pages).
> > I am not going to increase my already very high rating of this submission but I do believe that the quality has improved as a result of those efforts.
> > Please find below minor remarks.
> >
> > > Thanks! We have added a sentence in Sec. 3.1 to describe the node features in the main paper.
> >
> > Looks good, beware that there's a typo in "residUe types".
> >
> > > For angle prediction, since angular values are more sensitive to errors in protein structures than distances, we use discretized values for prediction.
> >
> > Surely, some reasonable amount of noise will keep most of the discretized values correct, but in the edge cases (for instance for angles close to the edges of the bins) where the discrete value is changed, errors will induce potentially more noise. So all in all, I am still not entirely convinced by this argument. But if the results are better with classification than with regression, that's fine by me.
> >
> > > During pretraining, we employ the proteins from AlphaFold Database containing only single-chain proteins. We admit that there may be some long proteins that contain structurally different motifs.
> >
> > Fair enough. I could imagine that especially after the more immediate benefits coming with the release of AlphaFold2 have been reaped, more emphasis will be put on modelling the proteins whose properties are harder to predict, such as multiple-chain proteins. If that is correct, I could imagine that this contrastive learning scheme might be problematic in such cases, but it makes sense to leave that for future work.

---

> ### Author Response · Authors · 2022-11-13
> **Author Feedback to Reviewer swLM (2/2)**
>
> >**Q6: I am not sure I really like the argument that the proposed method pre-trains with much less data than sequence-based approaches. While the number of proteins used for pre-training is indeed much lower, much more data per protein is being used when incorporating 3D structure.**
>
> Thanks for pointing out this. We have changed the phrase “with much less data” to “with much less pretraining data” in the abstract.
>
> >**Q7: Related to the previous point, it would be nice to see some runtime and memory usage analysis of the different methods used.**
>
> Table B: Running time and memory usage per epoch for ESM-1b and GearNet-Edge.
>
> |#Method|Running time|Memory|
> |:----:|:----:|:----:|
> |ESM-1b|5.3min|28.7G|
> |GearNet-Edge|4.5min|5.2G|
>
> This is a good point. But it is difficult to find a fair way to compare these pretraining methods, since the speed and memory cost highly depend on the computational resources and batch size used. All pretraining methods proposed in our paper can finish in 2 days on our cluster. Here we just report the costs for finetuning ESM-1b and GearNet-Edge per epoch in Table B, in which GearNet-Edge shows higher runtime and memory efficiency. It should be noted that we truncate the proteins up to 550 residues for ESM-1b to save memory while keeping the whole proteins for GearNet-Edge.
>
> >**Q8: Although it is qualitative, and not entirely compelling because the analysis has been performed only for the author's model, and it could be that they are hand picked, I find the results presented in Appendix H to be very interesting. It would be nice to at least put a reference to it in the main text.**
>
> We have put a reference at the beginning of Section 5 in the revised version.
>
> >**Q9: The current model includes mostly structural information about the backbone of the protein, which seems to be sufficient for the chosen downstream tasks. For tasks like complex interaction prediction, that probably will be a bottleneck. Do the authors have suggestions as to how to extend the current model to include side chain information?**
>
> Thanks for the question! We admit that side chain information is very important for complex tasks, e.g., protein-protein interaction, protein-ligand binding. In this work, we find that simply keeping CAs can achieve good performance on the considered tasks, since function and fold prediction only need coarser-level representations. Nevertheless, **our model can be easily generalized to protein side-chain modeling by extending the residue graphs to atom level**. Specifically, we can choose which atoms to keep for each protein (CA, backbones or side-chain atoms) and construct different types of edges between atoms (sequential, radius, knn or bond edges). In this way, we can explicitly model the interactions between atoms and thus enable the prediction for more complicated tasks, e.g., binding affinity prediction. It would be also interesting to explore different ways of constructing atom graphs, so we leave this for future work.

---

### Official Review · Reviewer_swLM · 2022-10-24

**Confidence:** 4
**Correctness:** 3
**Technical Novelty And Significance:** 3
**Empirical Novelty And Significance:** 3
**Recommendation:** 5

**Clarity, Quality, Novelty And Reproducibility:**

Clarity: Well-written, easy to follow.

Quality: Extensive experiment results.

Novelty: The proposed multi-view contrastive learning objective looks novel to me.

Reproducibility: Authors have claimed that all code and models will be released upon acceptance.

**Strength And Weaknesses:**

Pros:
1. The construction of protein graph with multiple edge types, each associated with a GCN kernel matrix is novel. Additionally, as shown in Table 3, there is a significant performance boost after introducing edge-type-specific convolution (GearNet-Edge vs. w/o relational convolution).
2. Different from commonly used self-supervised learning objectives that randomly masking out certain residues and/or coordinates, the proposed multi-view contrastive learning looks interesting and is reasonable to some extent.
3. The empirical evaluation is extensive, which includes four down-stream tasks and various network architectures and pre-trained models. The ablation study also reflects the importance of some key components of the proposed method.

Cons:
1. The proposed protein encoder only considers C-alpha atoms, which may be insufficient for certain down-stream tasks that require more fine-grained formulation of protein structures. For instance, to predict the binding affinity between protein and ligands, protein side-chain conformation should be considered.
2. The ablation study on the necessity of relation convolution in Table 3 is not entirely convincing. By replacing edge-type-specific convolutional kernels with a single shared one, the number of learnable model parameters is greatly reduced, which may limit the model capacity. How about a deeper network with shared convolutional kernel?
3. The proposed protein encoder is invariant to arbitrary translations, rotations, and reflections. I am not sure whether reflection-invariant is indeed necessary for protein representations since native L-amino-acid and unnatural D-amino-acid (only differ in the handedness) may have different physicochemical and biological properties.
4. For self-prediction baselines, the constructed protein graphs may have permanently lost detailed 3D coordinate information in the input node features, which may limit the self-prediction learning performance. How about those GNN models that explicitly consider node-wise 3D coordinates, e.g., EGNN?
5. In Table 3, it seems that “subspace + random edge masking” works better, even outperforming “GearNet-Edge (Multiview Contrast)”. So, why not directly using this for sampling sub-structures, rather than alternating between different sampling schemes?

**Summary Of The Paper:**

In this paper, authors propose a 3D structure-aware pre-training method for protein representation learning. Two novel structure-based protein encoders, GearNet and GearNet-Edge, are proposed to derive per-residue and whole-protein representations. The multi-view contrastive loss is adopted to maximize the similarity between different sub-structures of the same protein, while minimizing those from different proteins. Empirical evaluation is conducted on four down-stream tasks to verify the effectiveness of the proposed protein encoder and pre-training learning mechanism.

**Summary Of The Review:**

The overall quality is satisfying, but there are still a few unresolved concerns (as listed in the “Cons” section) that may need further discussion and clarification.

---

> ### Author Response · Authors · 2022-11-13
> **Author Feedback to Reviewer swLM**
>
> Thanks for your suggestions! Here is our response to your concerns.
>
> >**Q1: The encoder does not model side chain conformation.**
>
> Thanks for the question! We admit that side chain information is very important for complex tasks, e.g., protein-protein interaction, protein-ligand binding. In this work, we find that simply keeping CAs can achieve good performance on the considered tasks, since function and fold prediction only need coarser-level representations. Nevertheless, **our model can be easily generalized to protein side-chain modeling by extending the residue graphs to atom level**. Specifically, we can choose which atoms to keep for each protein (CA, backbones or side-chain atoms) and construct different types of edges between atoms (sequential, radius, knn or bond edges). In this way, we can explicitly model the interactions between atoms and thus enable the prediction for more complicated tasks, e.g., binding affinity prediction. It would be also interesting to explore different ways of constructing atom graphs, so we leave this for future work.
>
> >**Q2: The model capacity is limited for GearNet w/o relational convolution in the ablation study. How about a deeper network?**
>
> Thanks for the suggestion. In the revision version, for the baseline with a shared convolutional kernel, we run the model with different numbers of layers and report the number of parameters and performance in Table 3. The results show that **increasing the number of layers doesn’t make the baseline competitive with our model**. In contrast, simply using edge-type-specific convolutional kernels is an effective way to model proteins.
>
> >**Q3: Is reflection-invariant necessary for protein representations since native L-amino-acid and unnatural D-amino-acid (only differ in the handedness) may have different physicochemical and biological properties.**
>
> This is a good question! The **geometric invariance** mentioned in the paper refers to **the invariance with respect to the transformation on the whole protein structures** e.g., rotating a protein should not change its property. In contrast, **for local transformation on a residue, our model can capture the change** by distance and angular information and yield different outputs. Hence, our model can reflect the structural difference when replacing some L-amino-acids with D-amino-acids at the local residue level.
>
> >**Q4: For self-prediction baselines, the constructed protein graphs may have permanently lost detailed 3D coordinate information in the input node features, which may limit the self-prediction learning performance. How about those GNN models that explicitly consider node-wise 3D coordinates, e.g., EGNN?**
>
> As you said, our model discards detailed 3D coordinate information when constructing protein graphs. Instead, we model the spatial information via distance and angular information and also use these information as node and edge features. And it should be noted that though EGNN keeps 3D coordinates in the hidden layers, it still uses invariant features (i.e., distance) when injecting information to representations from these equivariant coordinates. Therefore, **the ways to utilize spatial information are similar in two models**. Our model further improves the capacity by incorporating relational and edge message passing mechanisms.
>
> Table A: Loss functions on AlphaFold Database when pretraining EGNN and GearNet-Edge with different methods.
>
> |#Method|Multiview Contrast|Residue Type Prediction|Distance Prediction|Angle Prediction|Dihedral Prediction|
> |:----:|:----:|:----:|:----:|:----:|:----:|
> |GearNet-Edge|0.472|1.207|0.026|0.344|1.175|
> |EGNN|2.128|2.338|1.120|1.376|1.574|
>
> In the latest version, we follow your suggestion to add experiments on pretraining EGNN in Appendix G. As shown in the paper, all pretraining methods work very well on EGNN. Among them, Distance Prediction and Multiview Contrast are the top two methods. As we observe **the pretraining loss on EGNN is much higher than that on GearNet-Edge** as shown in the table above, we think that **the limited capacity of EGNN may reduce the benefits from the pretraining method**. Therefore, it would be interesting to study how these pretraining methods perform on models with higher capacity so that we can fully exploit the potential of pretraining. We’d like to leave this as a future direction.
>
> >**Q5: Why not directly using this for sampling sub-structures, rather than alternating between different sampling schemes?**
>
> This is a good point. In the current version, we add more results on GO for this ablation study. According to Table 3, **the benefits of randomly sampling view functions in our method can be clearly observed on all three tasks of the GO dataset**. This can be understood since generating diverse view functions is very important for contrastive learning [a].
>
> [a] A simple framework for contrastive learning of visual representations. Chen et al., ICML, 2020.

---

> ### Author Response · Authors · 2022-12-09
> **Further suggestions are welcome**
>
> Thanks for your attention and response to our feedback. We are not sure if our feedback has addressed your concerns. We are happy to launch more discussions, and receive further suggestions to improve our paper. Please feel free to reach out if you have any question.

---

### Official Review · Reviewer_pcxa · 2022-11-03

**Confidence:** 4
**Correctness:** 4
**Technical Novelty And Significance:** 2
**Empirical Novelty And Significance:** 4
**Recommendation:** 6

**Clarity, Quality, Novelty And Reproducibility:**

The work is clear, and of good quality.  The methods used are not particularly novel.  The code and models are not currently provided, though the methods seem to be sufficiently described to be reproducible.

Minor typo: In section 4.1 the heading "contastive learning" is missing an 'r'.

**Strength And Weaknesses:**

The empirical results are compelling, demonstrating that the new wealth of structures can be successfully used to improve sequence-based models.  The proposed architecture is conceptually straightforward and does not include any particularly novel algorithmic contributions, though several components have not been used in the macromolecular representation learning area.  The ablation studies are a very nice addition as well.

**Summary Of The Paper:**

This work proposes a method to leverage unlabeled protein structural data to improve the prediction of various protein properties.  The authors present two main contributions: (1) a geometric encoder for protein structures, and (2) a multiview contrastive learning pretraining strategy, which together lead to state-of-the-art performance on several protein property prediction tasks.  The geometric encoder makes use of edge messaging passing, which the authors claim had not been used for macromolecular representation learning.  The use of different kinds of edges (sequential, radius, KNN) is also claimed to be novel for this task.

**Summary Of The Review:**

I recommend weak acceptance.  In my opinion, the empirical results and overall clarity of the paper outweigh the lack of algorithmic novelty, though the lack of code prevents better assessing reproducibility.

---

> ### Author Response · Authors · 2022-11-13
> **Author Feedback to Reviewer pcxa**
>
> Thanks for your suggestions! We have fixed the typo in the latest version. Here is our response to your concern.
>
> >**Q1: The methods used are not particularly novel.**
>
> The novelty of this work lies in the design of the proposed protein structure encoder and pretraining methods.
>
> Although there are many existing invariant encoders for molecules, their adaptation on proteins is non-trivial due to the size and different nature of proteins. In this work, we find that **simply constructing protein graphs with different edge types and modeling with different kernel matrices works very well**. Based on this novel observation, we propose a straightforward and effective encoder. This explains the novelty underlying our model and we believe the simpleness of our method will make it suitable for downstream applications.
>
> For pretraining methods, the idea of multiview contrastive learning has been commonly used in the machine learning community. Nevertheless, there is a lack of good view functions for proteins, which is very important to contrastive learning. In this work, **we design novel view functions that reflect protein substructures and show their effectiveness on downstream tasks**.

---

> ### Author Response · Authors · 2022-12-09
> **Further suggestions are welcome**
>
> Thanks for your attention and response to our feedback. We are not sure if our feedback has addressed your concerns. We are happy to launch more discussions, and receive further suggestions to improve our paper. Please feel free to reach out if you have any question.

---

### Author Response · Authors · 2022-11-13
**Summary of Response**

We would like to thank all reviewers for your time and constructive suggestions in our paper! We have improved the writing and experiments of the paper following your suggestions and marked the revised parts in red. Here is a brief summary of important points:

1. **The importance of randomly choosing different cropping and noise functions (Reviewer swLM, 86an):** This operation aims to generate more diverse views for contrastive learning. We have added experiments in Table 3 for demonstration.
2. **Pretraining on EGNN (Reviewer swLM):** We have added experiments with a different backbone model, EGNN, to prove the effectiveness of our proposed pretraining methods in Appendix G.
3. **Combine sequence- and structure-based encoders (Reviewer roE6):** To further show the advantage of incorporating structural information, we build structure-based models upon pretrained sequence representations. Experimental results in Appendix H show significant improvements over ESM-1b.
4. **Combine neural and retrieval-based methods (Reviewer roE6):** In Appendix I, we compare our methods with retrieval-based baselines on function prediction, fold classification and structure-based search tasks. We thoroughly discuss the pros and cons of neural and retrieval-based methods and show the potential of combining these two streams of methods with very strong results.

---

### Author Response · Authors · 2022-11-19
**Further suggestions are welcome**

Dear Reviewers,

Thank you for your review comments. Considering the deadline of Discussion Stage 1 is approaching, is there anything else we can do for our paper?

Thanks,

Authors

---

### Decision · Program_Chairs · 2023-01-20

**Decision:**

Accept: poster

**Justification For Why Not Higher Score:**

The overall work is largely building on existing methods, and as noted by the reviewers, while they have not quite be applied in macro-molecular settings, they are reasonably expected.

**Justification For Why Not Lower Score:**

The implementation and significant amount of testing and validation help to make this work a good study for acceptance, evident from the overall positive tone from all reviewers, and this work could help to inspire future work in this area given the importance of this topic.

**Metareview: Summary, Strengths And Weaknesses:**

This work propose a framework to leverage structure information to learn protein representations and a pre-training method via multi-view constrastive learning. Through comparison with a number of baselines including the more common sequence-based approaches, they demonstrate their proposed strategy could be useful for downstream tasks such as protein property classification, achieving state-of-the-art performance and could be synergistic with existing methods. I'd like to commend the authors for their diligent and careful work during the discussion stage that helped to improve the overall quality of the work. I do want to note that, it would be important for the authors to further note limitations of their work in a revised version of the manuscript based on the reviewers' responses and comments.

While the overall proposal is reasonably straightforward, the strengths of the paper are: (1) helps to bring attention to structural-based representation of macro-molecules; (2) showcase ideas that have not been widely used in protein representation learning / pre-training as noted by reviewers; (3) clear description and well-rounded experiments and evaluations after incorporating reviewers' suggestions.

There are still gaps that would need to be filled for future work, including how this work can be expanded both for more demanding tasks as noted by reviewer, such as the side chain representation, more demanding yet significantly more useful scenarios such as the representation and functional inferences of multi-domain, longer proteins that the authors have not quite addressed.

**Note From Pc:**

if the above contains the word "oral" or "spotlight" please see: "oral" presentation means -> notable-top-5% and "spotlight" means -> notable-top-25%. As stated in our emails, we are disassociating presentation type from AC recommendations